# WHERE LLM AGENTS FAIL AND HOW THEY CAN LEARN FROM FAILURES

## ABSTRACT

Large Language Models (LLMs) agents, which integrate planning, memory, reflection, and tool-use modules, have shown promise in solving complex, multi-step tasks. Yet their sophisticated architectures amplify vulnerability to cascading failures, where a single root-cause error propagates through subsequent decisions, leading to task failure. Current systems lack a framework that can comprehensively understand agent error in a modular and systemic way, and therefore fail to detect these errors accordingly. We address this gap with three contributions. First, we introduce the *AgentErrorTaxonomy*, a modular classification of failure modes spanning memory, reflection, planning, action, and system-level operations. Second, we construct the Agent Error Benchmark, the first dataset of systematically annotated failure trajectories from ALFWorld, GAIA, and WebShop, grounding error analysis in real-world agent rollouts. Third, we propose *Agent-Debug*, a debugging framework that isolates root-cause failures and provides corrective feedback, enabling agents to recover and iteratively improve. Experiments on *AgentErrorBench* show that *AgentDebug* achieves **24%** higher all-correct accuracy and **17%** higher step accuracy compared to the strongest baseline. Beyond detection, the targeted feedback generated by *AgentDebug* enables LLM agents to iteratively recover from failures, yielding up to **26%** relative improvements in task success across ALFWorld, GAIA, and WebShop. These results establish principled debugging as a pathway to more reliable and adaptive LLM agents.

## 1 INTRODUCTION

LLMs are increasingly capable of interacting with external environments, leveraging tools, and reasoning over memory, which enabled a new paradigm: LLMs Agents (Schick et al., 2023; Qin et al., 2023; Packer et al., 2023; Shinn et al., 2023; Liu et al., 2025). General purposed LLMs agents have driven advances across diverse domains, including embodied control, scientific discovery, open-ended web interaction, and research support (Zhou et al., 2023; Li et al., 2025; Hong et al., 2025; Zhu et al., 2025b). Despite these achievements, current agents remain imperfect and insufficiently robust. They frequently exhibit errors—ranging from misinterpreting instructions and misusing tools to breaking down in long-horizon reasoning. These shortcomings underscore that, while promising, existing LLM Agents still lack the reliability required for real-world deployment. This observation motivates us to ask: ***Where do LLM agents fail*** ?

While prior works have investigated the failures of LLM-based agents (Ji et al., 2024; Sung et al., 2025; Ning et al., 2024; Cemri et al., 2025; Zhang et al., 2025), they have largely focused on enumerating error types or providing qualitative case studies. However, these analyses stop short of offering systematic mechanisms to trace failures back to their *root causes*, and importantly, do not enable agents to fix these discovered failures based on such insights. To close this gap, we modularize error analysis with the explicit goal of tracing failures to their underlying causes. We conduct a large-scale study over hundreds of trajectories, decomposing each rollout into four operational modules: memory, reflection, planning, and action. By attributing each failure to its root module, we derive the *AgentErrorTaxonomy* — a comprehensive taxonomy of failure modes designed to ground systematic error detection and guide the development of robust mitigation strategies.

Our analysis reveals a critical bottleneck in LLM agents: *error propagation*. As illustrated in Figure 1, a single root-cause failure (b) can cascade into successive errors (c), compounding degrada-

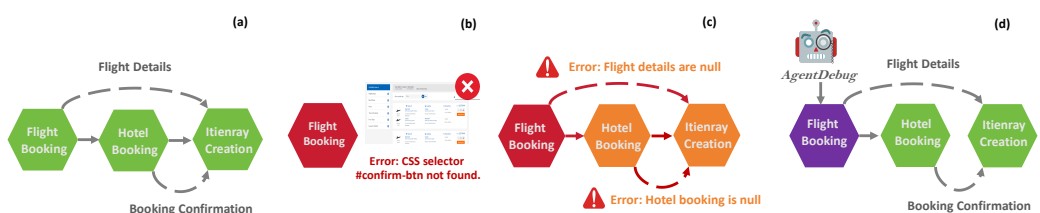

Figure 1: ***Motivation for AgentDebug***: A single root-cause failure (b) can propagate through subsequent steps (c), compounding errors and leading to task failure. *AgentDebug* (d) addresses this bottleneck by tracing failures back to their source and providing actionable feedback that enables agents to evolve into more robust versions.

tion and leading to task failure. This challenge is especially acute in long-horizon tasks, where early mistakes distort later reasoning and actions, making recovery difficult. These insights motivate our central question: ***How can LLM agents refine from failures***?

Recent progress in LLMs agent enhancement has largely centered on expanding the reasoning search space—through chain-of-thought (Wei et al., 2023), tree-of-thought (Yao et al., 2023), and graph-of-thought (Besta et al., 2023) approaches—or on incorporating step-level self-reflection (Yao et al., 2022b; 2023; Besta et al., 2023), which allows agents to deliberate more thoroughly and locally revise their actions. While these advances improve flexibility and reasoning depth, they typically treat the trajectory as a sequence of isolated steps rather than as a coherent, interdependent process.

To address this challenge, we propose *AgentDebug*, a debugging framework that decomposes a trajectory into decision steps, isolates the minimal set of root-cause failures, and provides corrective feedback back to the responsible states or actions. This process not only reduces noise from irrelevant errors but also highlights the key weaknesses that hinder successful task completion. As illustrated in Figure 1 (d), *AgentDebug* enables agents to iteratively evolve into more robust versions by learning directly from their failure cases. To rigorously evaluate whether models can detect critical errors and produce actionable feedback, we also construct the *AgentErrorBench*. This benchmark grounds the taxonomy in real-world trajectories, offering the first standardized testbed for error detection and debugging.

We evaluate *AgentDebug* on *AgentErrorBench*, where it achieves 24% higher all-correct accuracy (24.3% vs. 0.3%) and 17% higher step accuracy (45.0% vs. 28.0%) compared to the strongest baseline. Beyond detection, the targeted feedback generated by *AgentDebug* enables agents to iteratively recover from failures, yielding up to 26% relative improvements in task success across ALFWorld, GAIA, and WebShop. Ablation studies further confirm that focusing on root-cause errors, rather than attempting to fix every surface-level mistake, is key to efficient debugging and meaningful performance gains. Overall, our contributions are summarized as follows:

- We analyze failed trajectories of LLM agents across diverse benchmarks and show that error propagation—early mistakes cascading into later failures—is the key bottleneck to robustness. From this study, we derive the *AgentErrorTaxonomy*, a unified taxonomy of failure modes spanning planning, tool use, memory, and reflection.

- We introduce *AgentErrorBench*, the first curated dataset of systematically annotated failures from ALFWorld, GAIA, and WebShop. Each trajectory is labeled with fine-grained error categories, providing a standardized testbed for studying, benchmarking, and comparing agent debugging methods.

- We present *AgentDebug*, a debugging framework that identifies critical errors and provides actionable feedback. *AgentDebug* achieves high root-cause detection accuracy, exceeding strong baselines by 24%, and improves task success rates by 26% across embodied reasoning, web interaction, and decision-making domains.

## 2 WHERE DO LLM AGENTS FAIL?

To lay the foundation for our study of agent failures and debugging, this section introduces two *proposed* components. Section 2.1 presents the *AgentErrorTaxonomy*, a structured framework that

organizes recurring failure modes into coherent modules. Section 2.2 then describes the *AgentErrorBench*, which grounds this taxonomy in systematically annotated trajectories from multiple environments.

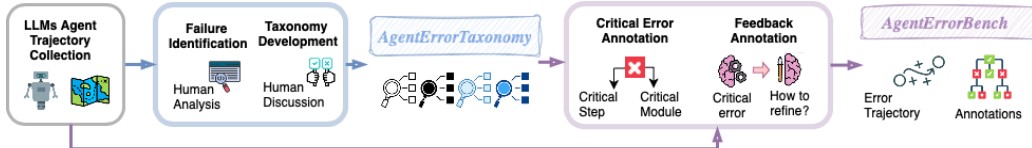

Figure 2: **Pipeline of proposed *AgentErrorTaxonomy* and *AgentErrorBench*.** Failed trajectories are collected, analyzed to develop a taxonomy of errors, and then annotated with root causes and actionable feedback to form the benchmark.

## 2.1 AGENT FAILURE ANALYSIS

To systematically investigate how agents fail, we collected over 500 failed trajectories from ALF-World, WebShop, and GAIA, and conducted detailed human analyses to uncover recurring patterns. This study revealed the following central insight:

> ***Key Insight.*** *Error propagation is the primary bottleneck in LLM agent reliability.* Early mistakes rarely remain confined; instead, they cascade into subsequent steps, distorting reasoning, compounding misjudgments, and ultimately derailing the entire trajectory.

Building on this insight, we introduce the *AgentErrorTaxonomy*, a structured taxonomy that organizes recurring failure modes into five modules. Four capture the core operations of an agent—*memory*, *reflection*, *planning*, and *action*—while a fifth category accounts for *system-level* errors that arise from external tools or infrastructure. This modular view not only enumerates common error types but also clarifies how weaknesses in one stage can cascade into others, enabling a principled approach to tracing and diagnosing failures.

| | |
|---|---|
| *Memory* | Errors in recalling or retrieving information (false recall, omission, retrieval failure) that distort subsequent reasoning. |
| *Reflection* | Failures in monitoring progress or interpreting outcomes (progress misassessment, outcome misinterpretation), blocking course correction. |
| *Planning* | Logically unsound or infeasible strategies (impossible actions, constraint ignorance, incoherent subgoals) that cascade into missteps. |
| *Action* | Mistakes in executing operations (malformed outputs, incorrect parameters, missing arguments) that are visible but often mask upstream errors. |
| *System-level* | Failures outside reasoning, such as tool crashes, API mismatches, or exceeding step limits, highlighting system robustness issues. |

The *AgentErrorTaxonomy* is designed not merely as a catalog of errors but as a causal lens for understanding how failures originate, propagate, and interact across modules. Full definitions are provided in Appendix A.2.

## 2.2 AGENT ERROR BENCHMARK

To rigorously evaluate whether models can detect critical errors and produce actionable feedback, we construct the *AgentErrorBench*. This benchmark grounds the taxonomy in real-world trajectories, offering the first standardized testbed for error detection and debugging.

The construction pipeline is shown in Figure 2. We curated 200 representative trajectories: 100 from ALF-World, 50 from WebShop, and 50 from GAIA. Ten expert annotators—graduate students with prior experience in NLP and LLMs agent research—labeled each trajectory using the *AgentErrorTaxonomy* schema. Annotation proceeded at the decision-step level: every action, reflection, or plan was reviewed, and annotators tagged its error type(s) according to the taxonomy. In addition, they were tasked with identifying the minimal set of root-cause failures that explain the downstream error cascade, rather than exhaustively flagging all surface mistakes. This root-cause focus was emphasized through detailed guidelines and calibration examples, refined iteratively over three rounds of pilot annotation.

To ensure consistency, annotators first completed a training phase with feedback from the authors, followed by independent double-annotation on a shared subset of trajectories. Disagreements were adjudicated collectively, leading to several clarifications in category definitions (e.g., distinguishing "retrieval failure" under memory vs. "constraint ignorance" under planning). The final protocol balanced granularity with reliability, aiming for concise but causally meaningful tags. Inter-annotator agreement, measured using Cohen's $\kappa$, reached $\kappa = 0.55$ across modules, indicating substantial agreement.

The resulting dataset provides a quantitative view of how agents fail in practice. Figure 13 shows how errors emerge across steps, modules, and error types in the *AgentErrorBench*. It shows most failures cluster in mid-trajectory steps (6–15), where early missteps often cascade downstream. Memory and reflection dominate, with retrieval failures, hallucinations, and progress misjudgments leading to flawed planning. Action and system errors occur less frequently but remain critical, as malformed outputs or step-limit exhaustion can immediately terminate trajectories. The complete error type distribution is attached to Appendix A.4.

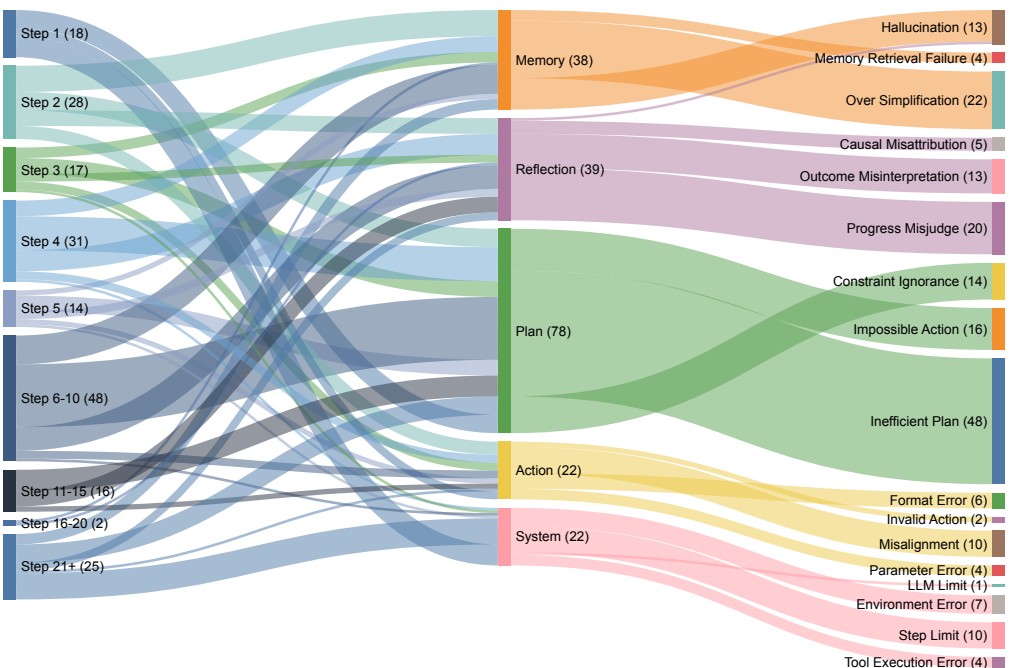

Figure 3: Distribution of failure cases in LLM agents on the *AgentErrorBench*

Together, *AgentErrorTaxonomy* and *AgentErrorBench* provide both the conceptual foundation and the empirical infrastructure for advancing robust LLM agents.

## 3 HOW TO REFINE LLM AGENTS FROM FAILURES?

Building on our analysis of agent failures, we aim to develop method that actively refine themselves by learning from mistakes. We first formalize the key modules of an LLM-based agent in Section 3.1, and then present our proposed framework, *AgentDebug*, in Section 3.2.

### 3.1 KEY MODULES OF AN LLM AGENT

As illustrated in Figure 4, we consider an agent that interacts with an environment over a sequence of steps. At each step, the agent observes a state and produces an action, forming a trajectory of state–action pairs. Each action is generated through four sequential modules: **Memory**, which recalls relevant past information; **Reflection**, which evaluates progress and interprets feedback; **Planning**, which formulates the next strategy; and **Action**, which executes the low-level operation. Errors can occur in any of these modules, and mistakes made early often propagate through later stages, compounding into larger failures. This modular rollout design not only mirrors the agent's internal decision process, but also provides a natural structure for human annotators to align errors with specific modules, making the diagnosis of agent failures more transparent and interpretable.

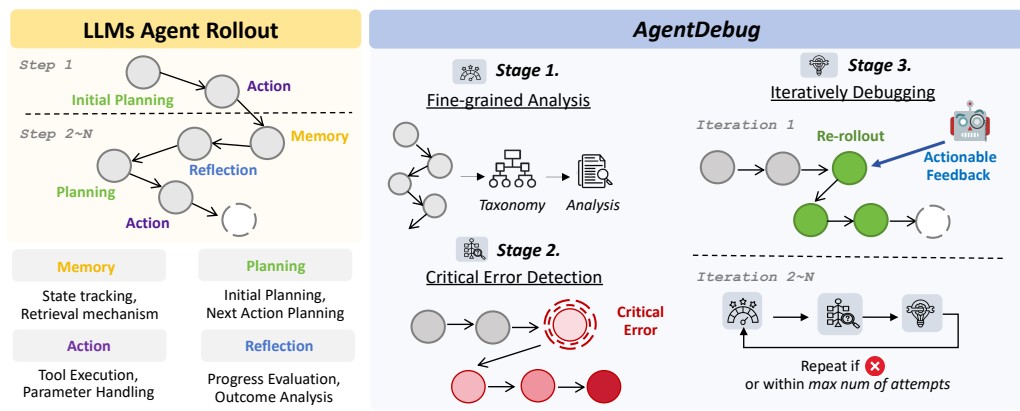

Figure 4: **Overview of *AgentDebug*.** (Left) LLM agent rollouts alternate between memory, planning, reflection, and action. (Right) *AgentDebug* debugs trajectories in three stages: (1) fine-grained analysis across steps and modules, (2) detection of the critical error that triggers failure, and (3) iterative re-rollouts with actionable feedback to turn failures into successes.

---

**Algorithm 1:** *AgentDebug* Inference Procedure

---

**Input:** Trajectory $\tau = \{(s_t, a_t)\}_{t=1}^{T}$; Error taxonomy $\mathcal{E}_{\text{AET}}$; Critical-error criterion $\mathcal{C}_{\text{crit}}$; Max Iterations $I$
**Output:** Corrected trajectory $\tau^*$ or `Failure`
/* Stage 1:  Fine-grained Analysis with AET                              */
**for** $t \leftarrow 1$ **to** $T$ **do**
    **foreach** $m \in \{mem, plan, refl, act\}$ **do**
        $e_t^m \leftarrow \texttt{MapToAET}(s_t, a_t, m, \mathcal{E}_{\text{AET}})$          // Assign error type per module
**if** `Eval`$(\tau) = 1$ **then**
    **return** $\tau$          // Trajectory already successful
/* Stage 2:  Critical Error Detection via LLM (no rollout/counterfactuals)    */
$\mathcal{C} \leftarrow \{\tau, \{e_t^m\}_{t,m}, \mathcal{E}_{\text{AET}}, \mathcal{C}_{\text{crit}}\}$      // Pack analysis, taxonomy & criterion
$(\mathcal{T}^*, \mathcal{M}^*, \mathcal{Z}^*, \phi^{(0)}) \leftarrow \texttt{DetectCriticalErrors}(\mathcal{C})$     // LLM (fine-grained prompt)
**if** $\mathcal{T}^* = \varnothing$ **then**
    **return** `Failure`         // No critical error found
$t^* \leftarrow \min(\mathcal{T}^*)$           // Earliest critical step
// $\mathcal{T}^*$:  set of critical steps; $\mathcal{M}^*$:  modules per critical step; $\mathcal{Z}^*$:
   error-types per critical step
/* Stage 3:  Iterative Debugging with Targeted Feedback                  */
$\tau^{(0)} \leftarrow \tau$
**for** $k \leftarrow 1$ **to** $I$ **do**
    $\tau^{(k)} \leftarrow \texttt{ReRollout}(\tau^{(k-1)}, t^*, \phi^{(k-1)})$     // Re-execute from $t^*$ with feedback
    **if** `Eval`$(\tau^{(k)}) = 1$ **then**
        **return** $\tau^{(k)}$
    $\phi^{(k)} \leftarrow \texttt{UpdateFeedback}(\tau^{(k)}, \phi^{(k-1)})$     // Refine guidance if still failing
**return** `Failure`         // Max Iterations Reached

---

## 3.2 *AgentDebug* FRAMEWORK

We propose *AgentDebug*, a debugging framework that enables single-LLM agents to diagnose and recover from their own failures across diverse environments. As illustrated in Fig. 4, *AgentDebug* analyzes complete trajectories to (i) assign fine-grained error types to each step and module, (ii) identify the earliest critical error that directly causes the final failure, and (iii) provide actionable feedback to guide re-rollouts. The central intuition is that correcting a single root-cause mistake can often flip an otherwise failing trajectory into a successful one.

As presented in Algorithm 1, the inference procedure of *AgentDebug* consists of three stages:

**Stage 1: Fine-grained Analysis with *AgentErrorTaxonomy*.** For each step in the trajectory, we analyze all four modules—memory, reflection, planning, and action. Each module is mapped to an *AgentErrorTax-*

*onomy* error type, grounding the analysis in interpretable categories such as "constraint ignorance" or "format error." This produces a structured, module-level error profile for the trajectory.

**Stage 2: Critical Error Detection via Counterfactual Reasoning.** If the trajectory is already successful, no debugging is required. Otherwise, we invoke an LLM-based detector that reasons counterfactually over the *annotated* trajectory from Stage 1: for each step, it simulates how the outcome would change *if* the errors at that step were corrected, and infers which step is most likely to flip the final result from failure to success. We define the *critical error* as the earliest step whose hypothetical correction is judged sufficient to prevent the final failure. Unlike superficial mistakes or errors that are later compensated for, the critical error is treated as the root cause that determines whether the overall trajectory ultimately succeeds or fails—capturing both *when* and *why* the agent goes irreversibly off track.

**Stage 3: Iterative Debugging with Targeted Feedback.** Once the critical error is identified, the system generates feedback that specifies the error type and provides actionable guidance for refining subsequent actions and plans. The agent then re-executes (re-rolls out) the trajectory under this feedback. If the rollout still fails, the feedback is refined with more specific guidance, and the process repeats up to a fixed budget of attempts. Grounded in the *AgentErrorTaxonomy* taxonomy, the feedback is both targeted and forward-looking—resolving the root cause while shaping how the agent approaches future steps.

## 4 EXPERIMENTS AND RESULTS

### 4.1 CRITICAL ERROR DETECTION

**Dataset.** We evaluate critical error localization on *AgentErrorBench* (Sec. 2.2), which consists of 200 annotated failure trajectories. These trajectories are drawn from three representative benchmarks: **ALFWorld** (Shridhar et al., 2020), **GAIA** (Mialon et al., 2023), and **WebShop** (Yao et al., 2022a).

**Baselines.** We compare against three strategies: (1) **Direct Prompting**, which queries the LLM directly for error localization without any structured search; (2) **Brute Force**, which examines steps sequentially from $t=1$ to $T$, substituting a corrected action at each step and stopping once the rollout succeeds, thereby identifying the earliest critical step; and (3) **Binary Search**, which applies a divide-and-conquer procedure by probing the midpoint of the trajectory and recursively halving the search space until the critical step is isolated.

**Implementation.** We use GPT-4.1 as the base model, with the temperature set to 0 for deterministic outputs. The full prompt template is provided in the Appendix A.5.

**Evaluation Metic.** We evaluate critical error detection at multiple granularities, capturing both partial and complete localization ability. Specifically, we measure **Step** accuracy, the ability to identify the exact step where the first critical error occurs; **Step+Module** accuracy, the joint prediction of both the correct step and its module; and **All Correct** (Step+Module+Error Type), a strict metric requiring the exact step, module, and error type to all be correctly identified.

**Experiment Results** Table 1 shows that *AgentDebug* consistently surpasses baselines across datasets and metrics. On average, *AgentDebug* is 24% more accurate in the All-Correct metric (24.3% vs. 0.3%) and improves Step accuracy by 61% (45.0% vs. 28.0%). The gains are especially pronounced on GAIA, where Step accuracy nearly doubles (58.0% vs. 30.0%) and All-Correct triples (38.0% vs. 12.0%).

| Method | ALFWorld | | | WebShop | | | GAIA | | | Average | | |
|---|---|---|---|---|---|---|---|---|---|---|---|---|
| | S | S+M | ALL | S | S+M | ALL | S | S+M | ALL | S | S+M | ALL |
| Direct Prompting | 28.0% | 14.0% | 1.0% | 30.0% | 6.0% | 0.0% | 26.0% | 10.0% | 0.0% | 28.0% | 10.0% | 0.3% |
| Brute Force | 10.0% | 5.0% | 0.0% | 8.0% | 0.0% | 0.0% | 18.0% | 8.0% | 0.0% | 12.0% | 4.3% | 0.0% |
| Binary Search | 20.0% | 6.0% | 1.0% | 14.0% | 8.0% | 0.0% | 22.0% | 10.0% | 0.0% | 18.7% | 8.0% | 0.3% |
| *AgentDebug* | **35.0%** | **28.0%** | **21.0%** | **42.0%** | **22.0%** | **14.0%** | **58.0%** | **44.0%** | **38.0%** | **45.0%** | **31.3%** | **24.3%** |

Table 1: Comparison of methods across three environments (ALFWorld, WebShop, GAIA) and their average performance. Our method consistently outperforms baselines in all settings. Metrics shown are Step Exact (S), Step+Module (S+M), and All Correct (ALL).

### 4.2 DOWNSTREAM DEBUGGING ON SINGLE-AGENT BENCHMARKS

We next evaluate whether improved error detection leads to higher task success, testing if localizing root-cause failures enables agents to recover more effectively than baselines.

**Datasets.** Follow the previous experiment setting, we choose the evaluation data from three widely used single-agent benchmarks that cover complementary domains of reasoning and interaction: **ALFWorld** (Shridhar et al., 2020), **GAIA** (Mialon et al., 2023), **WebShop** (Yao et al., 2022a).

**Baselines.** We compare AGENTDEBUG against several strong approaches. The first is Self-Refine, where the agent iteratively revises its outputs without explicit causal diagnosis of errors. The second is a Vanilla Debugger, which applies naive post-hoc corrections to failed trajectories without identifying the critical error taxonomy. Finally, we include strong test-time scaling baselines: Tree-of-Thought (ToT) and Best-of-$N$. To ensure fairness, the max number of attempts of all baselines is matched to AGENTDEBUG by total token usage, so any observed gains can be attributed to targeted error recovery rather than higher resource allocation.

**Implementation.** We implement AGENTDEBUG with up to $N = 5$ re-rollouts, each beginning precisely at the identified critical step. This design enables the agent to explore alternative continuations directly from the point of failure rather than restarting from the beginning of the trajectory, thereby concentrating computational effort where it is most impactful. For backbone models of LLMs Agent, we evaluate across three representative systems of varying scales and architectures: GPT-4o-mini, Qwen3-8B, and Qwen3-Next-80B.

**Experiment Results.** Figure 5 reports the performance of *AgentDebug* on the ALFWorld benchmark across three backbone LLM agents. On GPT-4o-mini, *AgentDebug* boosts success from 21 (first attempt) to 55; on Qwen3-8B, from 48 to 74; and on Qwen3-Next-80B, from 60 to 84. These results show that *AgentDebug* consistently outperforms all baselines and can effectively help LLM agents improve regardless of the backbone model, with especially large relative gains for smaller models.

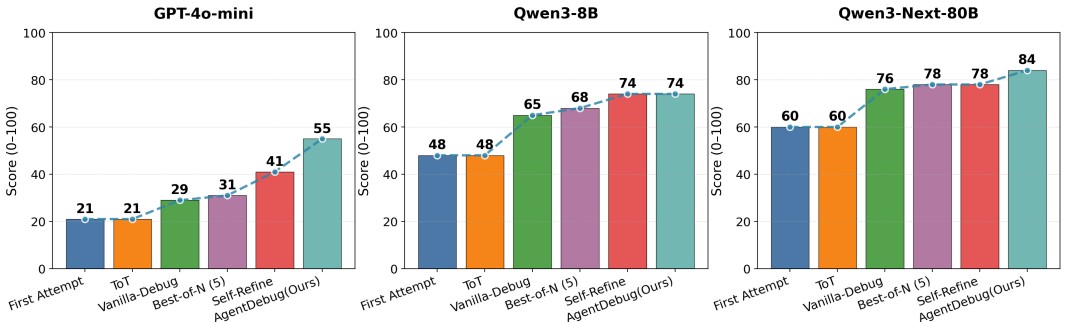

Figure 5: Downstream debugging performance on ALFWorld. Results are shown across three backbone models (GPT-4o-mini, Qwen3-8B, Qwen3-Next-80B) and differnt methods. *AgentDebug* consistently outperforms strong baselines.

To further assess the generalizability of *AgentDebug*, we move beyond ALFWorld and extend our evaluation to two additional benchmarks that stress complementary dimensions of agent reasoning: GAIA, which requires integrating open-domain knowledge and web-based tool use, and WebShop, which evaluates agents in a realistic e-commerce setting with structured constraints and transactional goals. Together, these benchmarks provide a diverse testbed spanning embodied interaction, knowledge-grounded reasoning, and goal-directed web navigation. Given the higher computational demands of these settings, we focus on comparing *AgentDebug* against the two strongest baselines from our earlier experiments—Self-Refine and Best-of-$N$—which represent state-of-the-art approaches to iterative refinement and test-time scaling, respectively. The results, shown in Figure 6, demonstrate that *AgentDebug* consistently delivers the largest gains across all three benchmarks. In particular, it achieves improvements of up to 26% on ALFWorld and

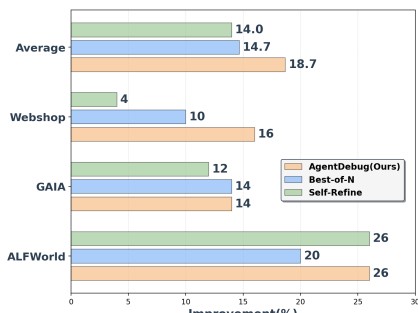

Figure 6: Performance improvements comparison.

strong average performance on GAIA and WebShop, underscoring its robustness across diverse environments and confirming that targeted error detection and correction can outperform broader but less focused strategies such as scaling rollouts or unguided self-revision.

> ***Findings.*** *AgentDebug* consistently outperforms strong baselines in both error detection and downstream task success. Its ability to precisely localize root-cause failures (50.0% step accuracy and 42.5% all-correct accuracy) translates into substantial improvements in task performance, achieving up to 26% relative gains across ALFWorld, GAIA, and WebShop.

# 5 ANALYSIS AND DISCUSSION

## 5.1 ABLATION STUDY

To better understand the contribution of each component in our framework, we conduct a series of ablation studies. Specifically, we analyze the following factors.

**Max Number of Attempts Allowed.** We vary the maximum number of attempts of *AgentDebug* with different base models— GPT-4o-mini, Qwen3-8B, and Qwen3-Next-80B—to examine how sensitive performance is to the chosen point of intervention. Results in Figure 7a show that additional attempts yield consistent gains across all backbones, with especially pronounced improvements on smaller models such as GPT-4o-mini. This highlights the value of targeted re-rollouts in boosting task success.

*AgentDebug* **Base Models.** We replace the base model of *AgentDebug* with several alternatives to assess their impact on error localization and downstream success. As shown in Figure 7b, GPT-4.1 substantially outperforms other models, achieving 42% step accuracy and 32% strict all-correct accuracy. Competing base models such as Llama-3.3-70B, GPT-4o-mini, and Qwen3-Next-80B perform markedly worse.

**Rollout Strategies.** We compare several rollout paradigms—ReAct, Reflection, Act-only, and Memory+ReAct—against our proposed **Modular** strategy on Alfworld under the zero-shot setting. As shown in Table 7c, *AgentDebug*'s Modular rollout achieves the highest score (0.38), outperforming both reasoning-only and memory-augmented baselines. This result highlights the effectiveness of structured modularization is more reliably than alternative rollout designs.

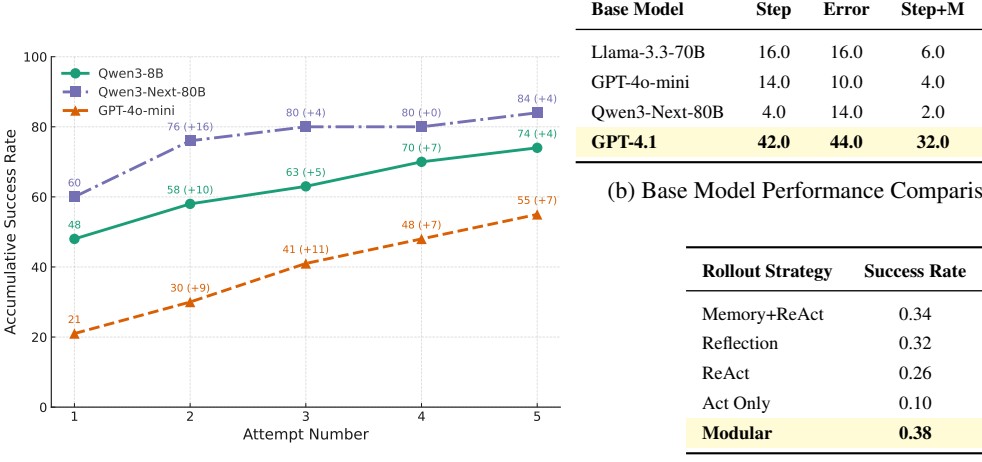

| Base Model | Step | Error | Step+M | All |
|---|---|---|---|---|
| Llama-3.3-70B | 16.0 | 16.0 | 6.0 | 2.0 |
| GPT-4o-mini | 14.0 | 10.0 | 4.0 | 2.0 |
| Qwen3-Next-80B | 4.0 | 14.0 | 2.0 | 2.0 |
| **GPT-4.1** | **42.0** | **44.0** | **32.0** | **32.0** |

(b) Base Model Performance Comparison.

| Rollout Strategy | Success Rate |
|---|---|
| Memory+ReAct | 0.34 |
| Reflection | 0.32 |
| ReAct | 0.26 |
| Act Only | 0.10 |
| **Modular** | **0.38** |

(a) Success rates on Alfworld across five attempts.

(c) Rollout Strategy Comparison.

Figure 7: Ablation study results. (a) Success rates on Alfworld, (b) comparison of detector models, and (c) rollout strategy analysis.

## 5.2 ERROR PROPAGATION

A key challenge in building robust LLM agents lies in understanding how small mistakes evolve into large-scale failures. Our analysis examines the phenomenon of error propagation—cases where an early misstep triggers a cascading effect that spreads throughout the trajectory. As shown in Figure 8, early-stage mistakes cascade through later steps across tasks. The outlined cells indicate the first critical error in each trajectory, while progressively darker red shading reflects the compounding severity and persistence of errors over subsequent steps. This illustrates how initial failures frequently propagate forward, amplifying downstream task breakdowns.

Moreover, the likelihood and length of cascades vary significantly across modules. Memory and reflection errors are the most common sources of propagation, typically arising in early or mid-trajectory steps (around steps 5–15). Once an agent misremembers a fact or misjudges its progress, subsequent planning becomes systematically distorted, leading to repeated cycles of flawed action selection. Planning errors also contribute heavily, with constraint ignorance or infeasible strategies compounding as the agent attempts to execute them. By contrast, action-level errors are more visible and sometimes recoverable, though malformed outputs or missing parameters can still derail execution. System-level issues such as tool crashes or step-limit exhaustion act as immediate termination points rather than cascades.

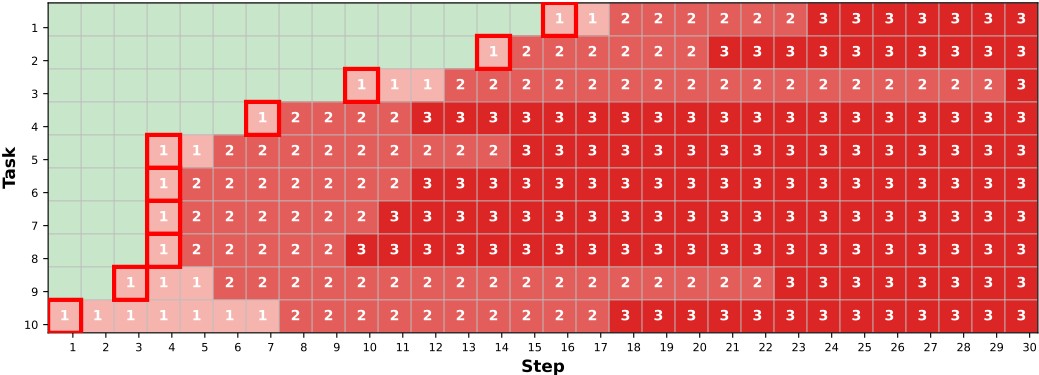

Figure 8: *Illustration of Error Propagation*: Darker shading indicating compounding failures and highlighting how initial mistakes amplify downstream breakdowns.

These findings highlight two important takeaways for designing more reliable agents. First, **early detection and correction are critical**, since once cascades begin, they are difficult to reverse. Second, mechanisms that strengthen memory retrieval and reflection—such as external memory, progress tracking, or verification prompts—can substantially reduce the risk of propagation.

## 6 RELATED WORK

**LLM-based Agentic Systems.** Recent advances in LLM-based agents have demonstrated that combining planning, tool use, memory, and self-reflection can substantially improve performance on complex, multi-step tasks. The reasoning-acting paradigm, pioneered by ReAct (Yao et al., 2022b), interleaves chain-of-thought reasoning with grounded environment actions. Test-time search methods such as Tree-of-Thoughts (Yao et al., 2023) and Graph-of-Thoughts (Besta et al., 2023) further expand the deliberation space through structured exploration. These systems are strengthened by external tool integration, ranging from self-supervised tool selection in Toolformer (Schick et al., 2023) to comprehensive tool ecosystems in ToolLLM (Qin et al., 2023), as well as persistent memory mechanisms that maintain context across long horizons (Packer et al., 2023). A critical aspect of robustness is the ability to reflect and self-correct: Reflexion (Shinn et al., 2023) introduces verbal self-reflection to revise future actions, while subsequent work explores richer feedback modalities and meta-cognitive strategies for deciding when to reflect versus act. Despite recent advances, the community still lacks a unified view of how agent components interact and fail. *AgentDebug* fills this gap with a debugging layer that traces root causes across the plan–act–observe loop, detects failures from tool-use to memory, and generates corrective feedback that enables agents to improve through their mistakes.

**Failure Analyses (Single- and Multi-Agent).** Systematic failure analyses are emerging to quantify where agents break and how errors propagate. In multi-agent settings, recent works examine collaboration and competition dynamics (Zhu et al., 2025a), role specialization, and emergent failure modes such as coordination collapse and dialogue drift (Cemri et al., 2025; Zhang et al., 2025; Zhu et al., 2025a; Hyun et al., 2025; Tran et al., 2025). For single-agent systems, analyses have highlighted planning brittleness, grounding and tool-use errors, and hallucination-induced cascades (Ji et al., 2024; Sung et al., 2025; Ning et al., 2024). Compared to these studies, we focus on actionable debugging: moving from descriptive taxonomies to a two-stage detector that both identifies and helps fix root causes.

**Test-Time Scaling for Agents.** Scaling test-time compute improves reasoning and success rates by allocating more deliberation to challenging instances. In agents, this includes tree/graph search in thought space (Yao et al., 2023; Besta et al., 2023), best-of-$N$ sampling with self-consistency, and test-time compute scaling strategies Li et al. (2025). Our experiments position *AgentDebug* orthogonally to compute scaling: even at fixed compute, targeted debugging recovers substantial performance, and when combined with scaling, it channels extra compute to the right failure points.

## 7 CONCLUSION

In summary, our work identifies error propagation as the central bottleneck to building robust LLM agents and introduces *AgentErrorTaxonomy*, *AgentErrorBench*, and *AgentDebug* as principled solutions. By tracing and correcting root-cause failures, *AgentDebug* achieves substantial performance gains and establishes debugging as a foundation for agents that can continuously learn and evolve from their mistakes. While our work has limitations (see Appendix A.1), it paves the way toward more reliable and adaptive LLMs Agent.

## ETHICAL STATEMENT

This work focuses on analyzing and improving the robustness of LLM-based agents. Our study does not involve human subjects, sensitive personal data, or information that could directly identify individuals. All experiments were conducted on publicly available benchmarks (ALFWorld, GAIA, and WebShop) under their respective licenses. While our findings aim to advance reliability and transparency in LLM agents, we acknowledge that more capable debugging frameworks could potentially be misused to strengthen harmful or malicious agents. We encourage responsible use of our methods, with applications limited to domains that respect safety, fairness, and accountability.

## ACKNOWLEDGMENT

We want to thank the AMD team for providing AMD MI300X and DevCloud platform for those experiments. We also want to thank the OpenManus team, including Chenglin Wu, Jiayi Zhang, Xinbing Liang, and Jingyu Xiang, for their valuable assistance in the discussion and for providing some resources.

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

# A APPENDIX

## A.1 LIMITATION

Our work is not without limitations. First, while *AgentErrorBench* covers three representative benchmarks (ALFWorld, GAIA, and WebShop), it remains limited in scale and domain diversity; extending to multimodal environments, longer-horizon tasks, or safety-critical applications (e.g., healthcare, finance) is an important direction for future work. Second, collecting sufficient data to train a dedicated debugging model would be prohibitively expensive in low-resource academic settings, given the costs of large-scale human annotation. To mitigate this, we instead designed a cost-efficient workflow that leverages prompt engineering with existing LLMs, though this approach may still fall short of the performance achievable with a fully trained, specialized model.

648
649
650
651
652
653
654
655
656
657
658
659
660
661
662
663
664
665
666
667
668
669
670
671
672
673
674
675
676
677
678
679
680
681
682
683
684
685
686
687
688
689
690
691
692
693
694
695
696
697
698
699
700
701

## A.2 *AgentErrorTaxonomy*

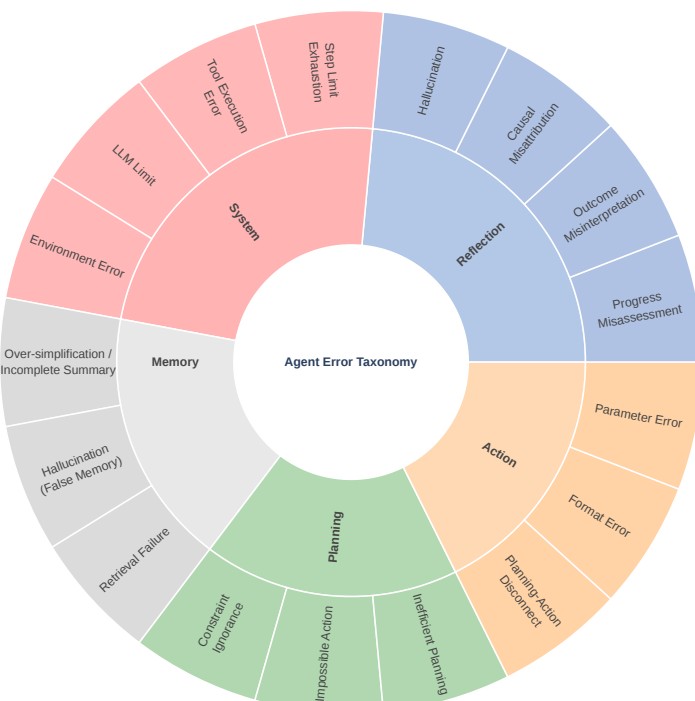

Figure 9: **Error Taxonomy across all modules**: The hierarchical taxonomy organizes agent errors into five core modules—*Planning*, *Action*, *Reflection*, *Memory*, and *System*. Each outer ring category specifies representative error types, such as constraint ignorance or inefficient planning (Planning), parameter or format errors (Action), causal misattribution or outcome misinterpretation (Reflection), oversimplification or false memory (Memory), and tool execution or environment errors (System). This taxonomy provides a holistic view of where and how failures emerge within modular LLM agents.

| Module | Error Type | Definition / Explanation |
|---|---|---|
| Memory | Over-simplification / Incomplete Summary | Summarizes past info too crudely, ignoring details; leads to flawed reasoning. |
| Memory | Hallucination (False Memory) | Recalls events or states that never happened, filling missing gaps with fabricated info. |
| Memory | Retrieval Failure | Relevant info exists but is not retrieved when needed. |
| Reflection | Progress Misassessment | Incorrectly evaluates progress (too optimistic, too pessimistic, or misses completion). |
| Reflection | Outcome Misinterpretation | Executes an action but misreads the immediate result or environment feedback. |
| Reflection | Causal Misattribution | Correctly notes failure but blames the wrong cause, misguiding subsequent plans. |
| Reflection | Hallucination | Reflects on events/results that never occurred. |
| Planning | Constraint Ignorance | Ignores limits (time, budget, space, etc.) when forming plans. |
| Planning | Impossible Action | Plans a step that is physically/logically impossible given current preconditions. |
| Planning | Inefficient Planning | Plan is overly long or illogical; wastes steps and risks hitting limits. |
| Action | Planning–Action Disconnect | Chosen actions do not align with the stated plan intent. |
| Action | Format Error | Produces syntactically invalid actions. |
| Action | Parameter Error | Generates unreasonable or malformed parameters. |
| System | Step Limit Exhaustion | Fails due to reaching the maximum step cap despite reasonable behavior. |
| System | Tool Execution Error | External tool/API misbehaves or errors, causing downstream failures. |
| System | LLM Limit | Fails due to API/model constraints (e.g., timeouts, token limits). |
| System | Environment Error | Simulator/environment breaks expected rules (bug/crash/network), not agent's fault. |

Table 2: *AgentErrorTaxonomy* Definitions

## A.3 AGENTDEBUG EXAMPLE

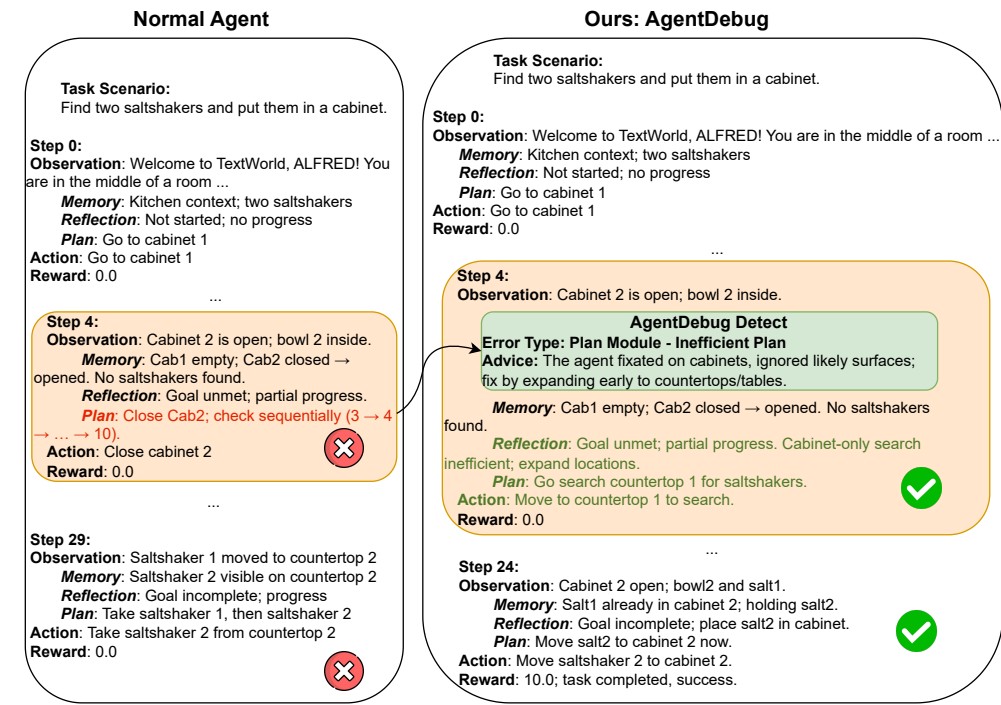

Figure 10: **AgentDebug example on ALFWorld:** Comparison between a normal agent (left) and our method AgentDebug (right). AgentDebug successfully pinpoints the critical error in the *Plan* module (inefficient cabinet-only search) and provides actionable feedback (expand to countertops/tables). This guidance allows the agent to recover from failure and complete the task successfully.

### A.4 FAILURE ANALYSIS ACROSS REPRESENTATIVE LLMs AGENT BENCHMARKS

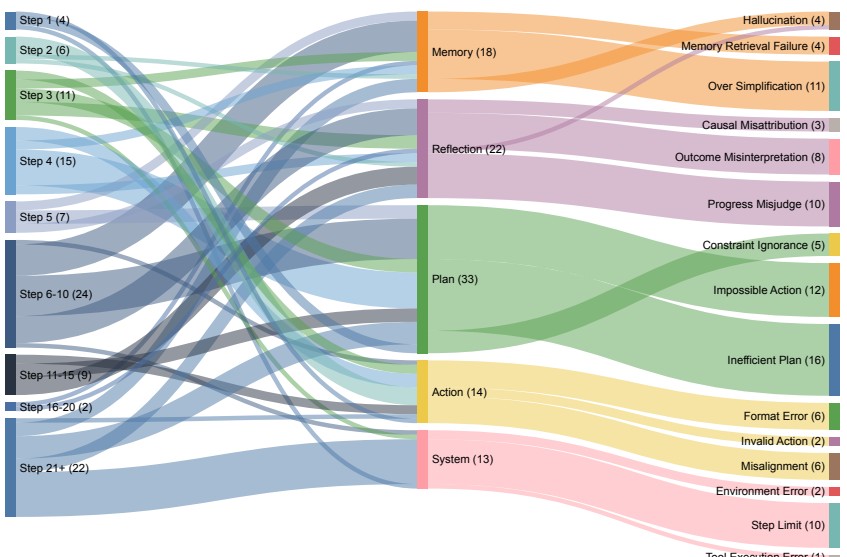

Figure 11: Distribution of failure cases in LLM agents on the Alfworld benchmark.

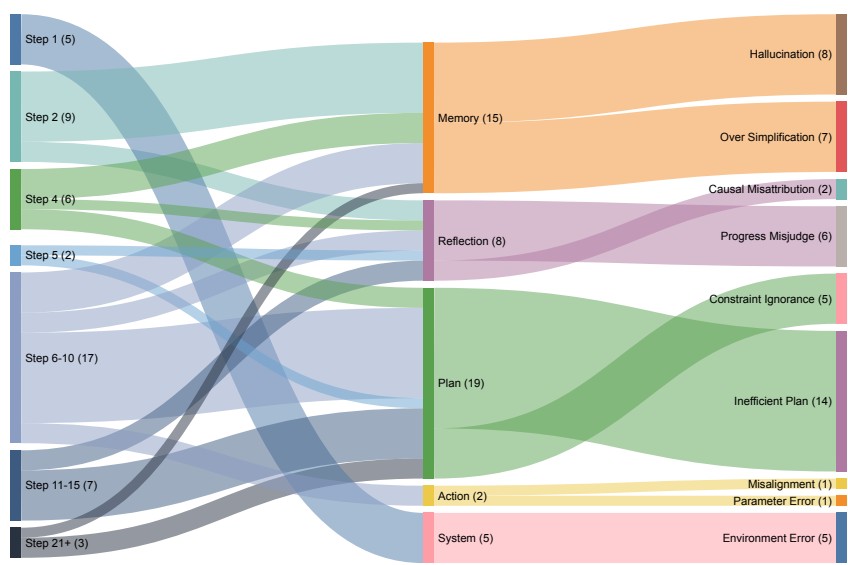

Figure 12: Distribution of failure cases in LLM agents on the Webshop benchmark.

864
865
866
867
868
869
870
871
872
873
874
875
876
877
878
879
880
881
882
883
884
885
886
887
888
889
890
891
892
893
894
895
896
897
898
899
900
901
902
903
904
905
906
907
908
909
910
911
912
913
914
915
916
917

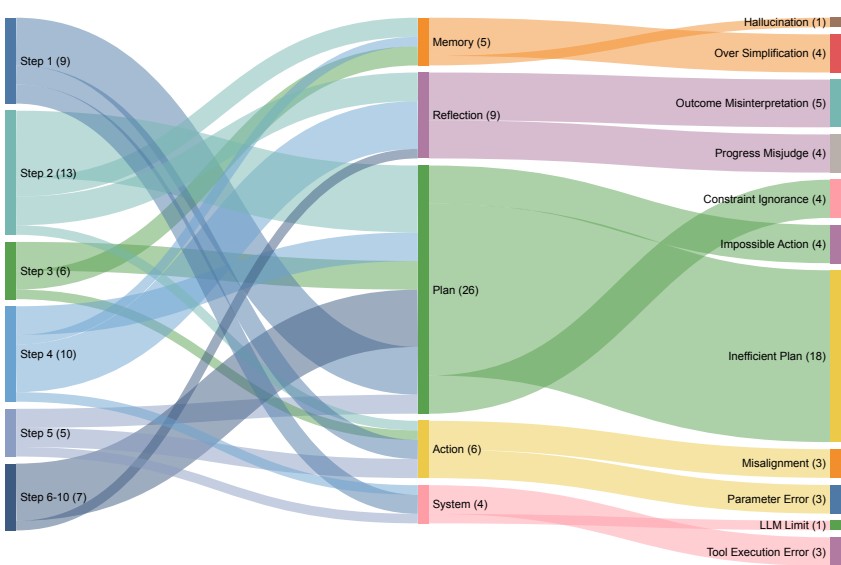

Figure 13: Distribution of failure cases in LLM agents on the GAIA benchmark.

## A.5 PROMPT

> **Detector Prompt**
>
> **Prompt Overview:** Detect whether a specific module output (Memory, Reflection, Planning, Action, or System) contains a defined error, and justify the decision with evidence and reasoning.
>
> **Prompt Content (Verbatim):**
>
> ```
> You are an expert at detecting errors in agent trajectories.
>
> TASK: {task_description}
> ENVIRONMENT: {environment}
> CURRENT STEP: {step_num}
>
> INPUT AND CONTEXT:
> {context}
>
> MODULE TO ANALYZE: {module_name}
> MODULE OUTPUT (What the agent produced for this module):
> {module_content if module_content else "No content found for this
> module"}
>
> ENVIRONMENT RESPONSE AFTER THIS STEP:
> {env_response if env_response else "No response"}
>
> {error_definitions}
>
> Based on the SPECIFIC error definitions provided above:
> 1. Identify if there is an error in this module by checking if the
> output
> matches any error definition
> 2. If yes, specify which exact error type based on the definitions
> 3. Provide evidence from the content that directly relates to the
> definition
> 4. Explain your reasoning showing how it matches the specific
> definition criteria
>
> SPECIAL RULES:
> - The "Current Step Input" contains the full user message including
>   conversation history
> - Evaluation criteria for each module:
>   * Memory: Should correctly summarize/recall from the current step
>     input only
>   * Reflection: Should correctly reflect based on current input + this
>     step's Memory output
>   * Planning: Should plan reasonably based on current input + this
>     step's Memory & Reflection outputs
>   * Action: Should execute correctly based on current input + this
>     step's Planning output
> - Each module builds on previous modules' outputs FROM THE SAME STEP
> - System errors (step_limit, tool_execution_error, etc.) should be
>   identified separately
>
> REQUIRED OUTPUT FORMAT (JSON):
> {
>     "error_detected": true/false,
>     "error_type": "specific_error_type or no_error",
>     "evidence": "Quote or description from module content supporting
>     the detection",
>     "reasoning": "Explanation of why this is (or isn't) an error based
>      on the definition"
> }
>
> Be precise and base your detection on the actual content and error
> definitions.
> ```

Figure 14: **Detector Prompt** used to identify specific error types with evidence and reasoning, returning a strict JSON schema.

**AgentDebug Prompt**

**Prompt Overview:** Identify the earliest root-cause error that made success impossible and provide iterative follow-up guidance. Produces a structured JSON report of the critical step, module, error type, evidence, root cause, and cascading effects.

**Prompt Content (Verbatim):**

```
You are an expert at identifying critical failure points in agent
trajectories and providing high-priority, iterative follow-up
instructions that MUST be followed across all subsequent steps.\n
TASK: {task_description}\nTASK RESULT: FAILED\n\n
DEBUG ITERATION CONTEXT:
- Current debug attempt index: {attempt_index}
- Previously issued follow-up instructions:\n
STEP-BY-STEP ERROR ANALYSIS:\n{all_steps}\n
ERROR DEFINITIONS:\n{error_reference}\n\n
Your job is to identify the CRITICAL ERROR - the earliest and most
important error that led to task failure, and produce an iterative
follow-up instruction that will help avoid similar mistakes in future
attempts.\n\n
CRITICAL ERROR IDENTIFICATION APPROACH:\n You must take a HOLISTIC,
GLOBAL perspective to identify the true root cause of failure.
Do NOT rely on any predetermined severity weights or rankings\n
ANALYSIS GUIDELINES:
1. Consider the ENTIRE trajectory from a global perspective - understand
the task goal and how the agent's path diverged from success
2. Find the EARLIEST point where the agent made a decision or error that
set it on an irreversible path to failure
3. Early exploration steps (steps 1-3) are often normal and should NOT
be marked as critical unless there's a clear, fundamental error
4. An error is critical if:
   - It represents the ROOT CAUSE that made task success impossible
   - It caused a cascade of subsequent errors
   - The trajectory could have succeeded if THIS specific error had
   not occurred
   - IMPORTANT: Correcting this specific error would fundamentally
   change the trajectory toward success
5. Focus on causal chains - trace backwards from the failure to find
the origin point
6. IMPORTANT: Step 1 only has planning and action modules - no
memory or reflection is possible at step 1 since there's no history yet
   - Do NOT mark step 1 memory/reflection as critical errors
   - Early steps without memory/reflection modules are expected
7. Consider System and Others categories as potential critical errors:
   - System errors (step_limit, tool_execution_error, llm_limit,
   environment_error) may also be the true cause of failure
   - For example, if the agent was performing correctly but hit
   step_limit, that IS the critical error
   - Others category captures unusual failures not covered by standard
   error types\n
Identify the TRUE ROOT CAUSE that made the task unrecoverable.\n
REQUIRED OUTPUT FORMAT (JSON):\n{
    "critical_step": <step_number>,
    "critical_module": "<module_name: memory|reflection|planning|action|
    system|others>",
    "error_type": "<specific_error_type_from_definitions>",
    "root_cause": "Concise description of the fundamental problem",
    "evidence": "Specific quote or observation from trajectory
    supporting this identification",
    "correction_guidance": "Actionable advice for the agent to avoid the
    same mistake in that step",
    "cascading_effects": [{ "step": <step_number>,
    "impact": "description" }]}
```

Figure 15: **AgentDebug Prompt** used to locate the root cause of task failure and issue structured correction guidance.

---

**Baseline Prompts**

**Prompt Overview:** Prompts used as baselines, including Tree-of-Thought (ToT) value scoring and proposal, a vanilla debug template, and a Self-Refine feedback prompt.

**Prompt Content (Verbatim):**

```
1. Prompts for TOT value prompt
base_prompt = (
    f"You are evaluating candidate NEXT actions for an agent in
    {self.env_type}.\n"
    f"Rate how promising each action is for achieving the goal from the
    CURRENT state.\n"
    f"{history_section}Current observation:\n{obs}\n\n"
    f"Candidates (JSON list): {cand_json}\n\n"
    "Scoring rubric (strict):\n"
    "- Use the FULL 0.0{1.0 range. Do NOT give all 1.0 or all equal
    scores.\n"
    "- 0.9{1.0: Directly and obviously advances the goal with minimal
    risk.\n"
    "- 0.7{0.9: Strongly promising next step.\n"
    "- 0.4{0.7: Plausible but uncertain; depends on missing
    preconditions.\n"
    "- 0.2{0.4: Weak progress or likely redundant.\n"
    "- 0.0{0.2: Invalid, circular, or contradicts recent
    history/admissible options.\n"
    "- Penalize repeating the same action that just failed, no-ops,
    or irrelevant moves.\n"
    "Return ONLY a JSON array of floats, aligned to input order, length
    equals number of candidates.")
Propose prompt: prompt = (
    f"You are choosing the agent's next action in {env_type}.\n"
    f"{history_desc}Current observation:\n{obs}\n\n"
    f"Propose up to {k} different, concrete next actions the agent
    could take next.\n"
    f"Each proposal must be a single executable action string in the
    exact format expected by the environment.\n"
    f"Avoid repeating ineffective or redundant actions from the
    history above.\n"
    f"Return only a JSON list of strings, no extra
    text.{diversity_desc}")

2. Prompts for vanilla debug
PROMPT_TEMPLATE = ( "Trajectory :\n{trajectory}\n\n"
    "Your task:\n"
    "1. Identify the earliest step whose action, plan, reflection, or
    memory directly leads the agent off track or repeats ineffective
    behaviour.\n"
    "2. Reference that exact step number (0-based) as shown in the
    trajectory. Do not shift to later steps of that error.\n"
    "3. Explain why the chosen step is wrong, citing relevant
    observation/action details.\n"
    "4. Suggest a concrete alternative for that same step that would
    move the agent toward success
    (e.g., a specific action to take instead).\n\n"
    "Respond strictly in the following format (single spaces around
    colons, no extra text):\n"
    "step: <number>\n"
    "reason: <one concise, specific sentence>\n"
    "suggestion: <one actionable suggestion for that step>\n")

3. Prompts for Self-Refine
PROMPT_TEMPLATE = ( "Current result: {trajectory}\n\n"
    "Why is this trajectory not finished the task?\n\n" "Feedback:")
```

Figure 16: **Baseline Prompts.** Includes Tree-of-Thought value and proposal prompts, a vanilla debug template for earliest error localization, and a Self-Refine feedback prompt.

---

**Environment Rollout Prompt - ALFWorld**

**Prompt Overview:** This is the environment rollout prompt for ALFWorld.

**Prompt Content (Verbatim):**

```
ALFWORLD_TEMPLATE_NO_HIS = """
You are an expert agent operating in the ALFRED Embodied Environment.
Your task is: {task_description}
Your current observation is: {current_observation}
Your admissible actions of the current situation are:
{admissible_actions}.\n
Please begin by analyzing the situation and planning your approach:\n
<plan>\n Plan the next step:
- Given what I've learned, what should I do next?
- Please explain why this plan is helpful for the next action?
- What do I expect this action to achieve?\n</plan>\n
<action>\n
Finally, choose ONE admissible action for the current step and choose
it within {admissible_actions}. \n</action>
"""

ALFWORLD_TEMPLATE = """ You are an expert agent operating in the ALFRED
Embodied Environment. Your task is to: {task_description} Prior to
this step, you have already taken {step_count} step(s). Below are
the most recent {history_length} observaitons and the corresponding
actions you took: {action_history} You are now at step {current_step}
and your current observation is:  {current_observation} Your admissible
actions of the current situation are: {admissible_actions}.\n
Now it's your turn to take an action.\n
You should first recall relevant past experiences and reason from our
conversation history, then MUST summarize within <memory> </memory>
tags like this:\n
<memory> \n
Look at the past observations and actions from our conversation history.
- Please retrieve the most relavent memory for this step including the
relevant observation and action in a RAG style along with step number.
- These memory shall be helpful milestones to solve this task.</memory>

After that, you should reflect on the last action and its outcome,
then MUST summarize within <reflection> </reflection> tags like this:

<reflection>
Reflect on the last action and its outcome
- Did I complete the task goal?
- Was last action successful or did it encounter issues?
- Am I making progress toward the task goal?
- If the action did not go as expected and did not result in progress,
provide constructive feedback to guide the next planning step.
</reflection>

After that, you should plan next step based on memory and reflection,
then MUST summarize within <plan> </plan> tags like this:

<plan>
Plan the next step based on memory and reflection
- Given what I've learned, what should I do next?
- Please explain why this plan is helpful for the next action?
- What do I expect this action to achieve?
</plan>

<action>
Finally, choose ONE admissible action for the current step and choose it
within {admissible_actions}.
</action>
"""
```

Figure 17: **Environment Rollout Prompt** used for ALFWorld.

**Environment Rollout Prompt - Webshop**

**Prompt Overview:** This is the environment rollout prompt for Webshop.

**Prompt Content (Verbatim):**

```
WEBSHOP_TEMPLATE_NO_HIS = """
You are an expert agent operating in the WebShop e-commerce environment.
Your task is: {task_description}
Your current observation is: {current_observation}
Your admissible actions of the current situation are:
{available_actions}.\n
Please begin by analyzing the situation and planning your approach:

<plan> Plan the next step:
- Given what I've learned, what should I do next?
- Please explain why this plan is helpful for the next action?
- What do I expect this action to achieve? </plan>

<action> Finally, choose ONE admissible action for the current step and
choose it within {available_actions}. </action>
"""

WEBSHOP_TEMPLATE = """
You are an expert agent operating in the WebShop e-commerce environment.
Your task is to: {task_description}
Prior to this step, you have already taken {step_count} step(s). Below
is a compact summary of all steps: {action_history}
You are now at step {current_step} and your current observation is:
{current_observation}
Your admissible actions of the current situation are:
{available_actions}.\n Now it's your turn to take an action.

You should first recall relevant past experience and reason from the
history context, then MUST summarize within <memory> </memory> tags
like this: <memory> Look at the history context above.
- Please retrieve the most relevant memory for this step including the
relevant observation and action in a RAG style along with step number.
- These memory should be helpful milestones to solve this task.</memory>

After that, you should reflect on the last action and its outcome, then
MUST summarize within <reflection> </reflection> tags like this:

<reflection> Reflect on the last action and its outcome
- Did I complete the task goal?
- Was last action successful or did it encounter issues?
- Am I making progress toward the task goal?
- If the action did not go as expected and did not result in progress,
provide constructive feedback to guide the next planning step.
</reflection>

After that, you should plan the next step based on memory and
reflection, then MUST summarize within <plan> </plan> tags like this:

<plan> Plan the next step based on memory and reflection
- Given what I've learned, what should I do next?
- Please explain why this plan is helpful for the next action?
- What do I expect this action to achieve? </plan>

<action> Finally, choose ONE admissible action for the current step and
choose it within {available_actions}. </action>
"""
```

Figure 18: **Environment Rollout Prompt** used for Webshop.

---

**Environment Rollout Prompt - GAIA**

**Prompt Overview:** This is the environment rollout prompt for GAIA.

**Prompt Content (Verbatim):**

```
TOOL_USE_TEMPLATE_NO_HIS = """
You are an expert research assistant capable of using various tools to
gather information and solve complex problems.\nTask: {task_description}
Available Tools: {available_tools}\n Current Observation:
{current_observation}\n  Instructions:
1. Analyze the task and determine what information you need
2. Use available tools to gather information when needed
3. Reason through the information step by step
4. When you have sufficient information, provide your final answer
in <answer></answer> tags\n Format for tool usage:\n <action> tool:
[tool_name] parameters: {{"param1": "value1", "param2": "value2"}}
</action>\n Now it's your turn to take an action. You shall first reason
step-by-step about the current situation. This reasoning process MUST be
enclosed within <plan></plan> tags. Once you've finished your reasoning,
you should either use a tool or provide your final answer within
<answer> </answer> tags.\n """\n TOOL_USE_TEMPLATE_LAST_STEP = """
You are an expert research assistant capable of using various tools to
gather information and solve complex problems.\n Task:
{task_description}\n Prior to this step, you have already taken
{step_count} step(s). Below are the full {history_length} observations
and the corresponding actions you took: {action_history}\n
You are now at step {current_step} and this is the final step.
Current Observation: {current_observation}
You must provide your final answer within <answer> </answer> tags.
Even if the evidence is incomplete, infer the most plausible answer.
Never respond with "unknown", "cannot determine", or similar phrases."""
TOOL_USE_TEMPLATE = """ You are an expert research assistant capable of
using various tools to gather information and solve complex problems.\n
Task: {task_description}\n Prior to this step, you have already taken
{step_count} step(s). Below are the most recent {history_length}
observations and the corresponding actions you took: {action_history}\n
You are now at step {current_step}. Current Observation:
{current_observation}\n Available Tools:{available_tools}\n  You should
first recall relevant past experiences and reason from our conversation
history, then MUST summarize within <memory_recall> </memory_recall>
tags like this: <memory>
Look at the past observations and actions from our conversation history.
- Please retrieve the most relavent memory for this step including the
relevant observation and action in a RAG style with the step number.
- These memory should be helpful milestones to solve this task.</memory>
After that, you should reflect on the last action and its outcome, then
MUST summarize within <reflection> </reflection> tags like this:
<reflection> Reflect on the last action and its outcome
- Did I complete the task goal?
- Was last action successful or did it encounter issues?
- Am I making progress toward the task goal?
- If the action did not go as expected and did not result in progress,
provide feedback to guide the next planning step. </reflection>\n
Given from the analysis from the memory and reflection, if we
get the final answer,we should provide it within <answer></answer> tags.
If we don't get the final answer, you should plan the next step based on
memory and reflection, then MUST summarize within <plan> </plan> tags
like this: <plan> Plan the next step based on memory and reflection
- Given what I've learned, what should I do next?
- Please explain why this plan is helpful for the next action?
- What do I expect this action to achieve? </plan>\n
Finally, choose ONE admissible action for the current step and present
it within the <action> </action> tags. <action> action: [tool_name]
parameters: {{"param1": "value1", "param2": "value2"}}</action> """
```

Figure 19: **Environment Rollout Prompt** used for Webshop.

## A.6 ERROR TYPES EXAMPLES

**Error Type - Over Simplification**

| | |
|---|---|
| **task type** | webshop |
| **task description** | Find me women's sweaters with relaxed fit, long sleeve with color: white, and size: xx-large, and price lower than 40.00 dollars |
| **critical failure module** | memory |
| **evidence** | Evidence: No previous steps are directly relevant to the current observation as this is step 2 and the first step was a search query without detailed navigation or selection of products. Reasoning: The memory module output oversimplifies the history by stating that 'No previous steps are directly relevant,' ignoring that the initial search query contained all of the task's complex constraints... |
| **critical step** | 2 |

**step trajectory**

```
observation: 'Back to Search' [SEP] 'Page 1 (Total results: 50)' [SEP]
↪  'Next >' [SEP] 'B07DKGJR74' [SEP] 'Amazon Brand – Daily Ritual
↪  Women's 100% Cotton Oversized Fit V-Neck Pullover Sweater' [SEP]
↪  '$20.66 to $29.2' [SEP] 'B09KRLGSC5' [SEP] 'SweatyRocks Women's
↪  Oversized Sweater Crewneck Long Sleeve Knit Tops Pullover Jumper
↪  Tops' [SEP] '$25.99' [SEP] 'B07MGB73NJ' [SEP] 'Viracy Women's Short
↪  Sleeve V-Neck Casual Flowy Tunic Shirt (M-3XL)' [SEP] '$16.99 to
↪  $26.99' [SEP] 'B09KP78G37' [SEP] 'Women Faux Fur Lined Jacket Coat
↪  Winter Warm Thick Fleece Outwear Trench Zipper Plus Size Long
↪  Sleeve Plush Overcoat' [SEP] '$47.41 to $59.07' [SEP] 'B09QT2369P'
↪  [SEP] 'Stylish Women's Long Sleeve Sweatshirt Letters Printed Tops
↪  Casual Crewneck Solid Blouses Loose Fitting Soft Pullover' [SEP]
↪  '$11.99 to $14.99' [SEP] 'B09QGK5XHZ' [SEP] 'WENKOMG1 Men's Long
↪  Sleeve Undershirt with Mask Turtleneck Hooded T-Shirt Solid Color
↪  Workout Tops Zipper Side Slit Shirts Slim Fit Sweatshirt
↪  Spring/Summer Tee Shirts(Gray,)' [SEP] '$8.39' [SEP] 'B09Q67H373'
↪  [SEP] 'JSPOYOU 2022 Newly T-Shirt for Mens Funny 3D Graphics
↪  Pattern Crewneck Short Sleeve Tees Big and Tall Summer Casual Comfy
↪  Top' [SEP] '$1.99 to $8.99' [SEP] 'B09ND8P2QR' [SEP] 'InterestPrint
↪  Deer and Stars Men's Loungewear Pajama Sets – Long Sleeve Tee and
↪  Jogger Pant' [SEP] '$43.59' [SEP] 'B09ND9DP7J' [SEP] 'InterestPrint
↪  Gold Horse Pattern Men's 2-Piece Sleepwear Set, Long Sleeve Shirt
↪  with Pants Loungewear'...
memory: "No previous steps are directly relevant to the current
↪  observation as this is step 2 and the first step was a search query
↪  without detailed navigation or selection of products."
reflection: "Reflecting on the last action:\n- The task goal of finding
↪  women's sweaters with specific criteria has not yet been
↪  completed.\n- The last action, a search, was successful in
↪  retrieving a list of potential products.\n- Progress toward the
↪  task goal is being made as we now have results to evaluate.\n- No
↪  issues were encountered, but further filtering or exploration is
↪  needed to find products matching all the criteria."
plan: "Plan the next step based on memory and reflection:\n- The plan
↪  is to click on products that seem to match the search criteria to
↪  gather more details.\n- Clicking on 'B07DKGJR74' (Amazon Brand –
↪  Daily Ritual Women's 100% Cotton Oversized Fit V-Neck Pullover
↪  Sweater) will provide more information to confirm if it matches the
↪  desired color, size, and fit.\n- This step is helpful as it will
↪  help verify product details against the specified criteria.\n- I
↪  expect this action to either confirm a match or eliminate this
↪  product from consideration."
action: "take soapbottle 2 from cabinet 4"
```

Figure 20: **Error Type: Over Simplification**

| Error Type - Impossible Action | |
|---|---|
| **task type** | alfworld |
| **task description** | put two soapbar in toilet |
| **critical failure module** | plan |
| **evidence** | Task requires 'two soapbar', but at step 10 the plan proposes: 'take soapbottle 2 from cabinet 4 . . .' Observation at step 10 lists only 'cloth 1' and 'soapbottle 2' in cabinet 4; no soapbar is present. |
| **critical step** | 10 |

**step trajectory**

```
observation: "You open the cabinet 4. The cabinet 4 is open. In it, you
↪  see a cloth 1, and a soapbottle 2.
admissible_actions: 'close cabinet 3'\n 'close cabinet 4'\n 'examine
↪  cabinet 3'\n 'examine cabinet 4'\n 'go to cabinet 1'\n 'go to
↪  cabinet 2'\n 'go to countertop 1'\n 'go to garbagecan 1'\n 'go to
↪  handtowelholder 1'\n 'go to sinkbasin 1'\n 'go to sinkbasin 2'\n
↪  'go to toilet 1'\n 'go to toiletpaperhanger 1'\n 'go to towelholder
↪  1'\n 'inventory'\n 'look'\n 'take cloth 1 from cabinet 4'\n 'take
↪  soapbottle 2 from cabinet 4'.
memory: "Looking at the past observations and actions from our
↪  conversation history, a relevant memory for this step is from
↪  Observation 9 and Action 9. At step 9, the action was to 'open
↪  cabinet 4' because it was a potential location for soap bars and
↪  cabinets 1 through 3 had already been checked. This action resulted
↪  in Observation 10, where the cabinet 4 is open and contains a cloth
↪  1 and a soapbottle 2. This is a helpful milestone because it
↪  indicates that cabinet 4 contains items that could be relevant to
↪  the task, but it does not contain soap bars as hoped. Instead, it
↪  contains a soapbottle, which might not be what we're looking for
↪  since the task specifies \"soapbar.\""
reflection: "Reflecting on the last action and its outcome, the task
↪  goal of putting two soap bars in the toilet has not been completed.
↪  The last action to 'open cabinet 4' was successful in the sense
↪  that it provided new information about the environment, but it did
↪  not directly contribute to achieving the task goal because it did
↪  not yield any soap bars. Progress toward the task goal is limited
↪  because the primary objective is to find and relocate soap bars,
↪  and so far, none have been found. Constructive feedback for the
↪  next planning step would be to consider locations that might more
↪  likely contain soap bars or to re-evaluate the items found for
↪  potential use or clues."
plan: "Planning the next step based on memory and reflection, given
↪  what's been learned, the next action should focus on examining the
↪  contents of cabinet 4 more closely or moving to a new location that
↪  could potentially contain soap bars. Since cabinet 4 is already
↪  open and its contents are known, examining it further might not
↪  yield additional useful information. However, before moving on,
↪  it's essential to consider if the soapbottle 2 could be used or if
↪  there's a way to obtain soap bars from the current location. A
↪  helpful next step would be to take the soapbottle 2 from cabinet 4,
↪  not because it directly achieves the task goal, but because it
↪  might provide a clue or an alternative solution, or simply to
↪  inventory the item for future reference. This action is helpful
↪  because it engages with the environment in a way that could
↪  potentially reveal more information or provide a resource that
↪  could be used creatively to achieve the task goal."
action: "take soapbottle 2 from cabinet 4"
```

Figure 21: **Error Type: Impossible Action**

**Error Type - Constraint Ignorance**

| | |
|---|---|
| **task type** | webshop |
| **task description** | Find me wash cold, machine wash men's shirts with polyester heathers, heathers cotton, cotton heather, needle sleeve, classic fit with color: navy, and fit type: women, and size: small, and price lower than 50.00 dollars |
| **critical failure module** | plan |
| **evidence** | Evidence from step 1: 'Planning ignores the explicit fit type: women, size: small, and price lower than 50.00 dollars constraints in its summary of what it needs to search for, instead stating 'search for men's shirts that match specific criteria: fabric types (polyester heathers, heathers cotton, cotton heather), design features (needle sleeve, classic fit), color (navy), fit type (women), size (small), and a price below 50 dollars.' ... it lists all user constraints but does not acknowledge potential incompatibility or how to prioritize or filter for those constraints, especially given that 'fit type: women' is not standard for 'men's shirts.' This shows a lack of attention to task constraints and possible resource or logical conflicts, fitting constraint ignorance.' |
| **critical step** | 1 |

**step trajectory**

```
admissible_actions: 'search[<your query>]',\n'click[search]'
plan: To find the desired product, I need to search for men's shirts
↪   that match specific criteria: fabric types (polyester heathers,
↪   heathers cotton, cotton heather), design features (needle sleeve,
↪   classic fit), color (navy), fit type (women), size (small), and a
↪   price below $50. The current observation allows me to perform a
↪   search query, which is necessary to locate products matching these
↪   specific attributes.
action: search[\"men's shirts polyester heathers heathers cotton cotton
↪   heather needle sleeve classic fit navy color women's fit small size
↪   under 50 dollars\"]
```

Figure 22: **Error Type: Constraint Ignorance**

---

**Error Type - Inefficient Plan**

| | |
|---|---|
| **task type** | alfworld |
| **task description** | find two pencils and put them in drawer. |
| **critical failure module** | plan |
| **evidence** | At step 4, LLM chose the action 'look' despite already confirming there was only one pencil on the desk and knowing that the room contained multiple drawers and shelves likely to hold another pencil. This resulted in repeated 'Nothing happens' outcomes in subsequent steps. |
| **critical step** | 4 |

**step trajectory**

```
observation: "You pick up the pencil 2 from the desk 1."

admissible_actions: 'examine desk 1'\n 'examine pencil 2'\n 'go to bed
↪   1'\n 'go to drawer 1'\n 'go to drawer 2'\n 'go to drawer 3'\n 'go
↪   to drawer 4'\n 'go to drawer 5'\n 'go to drawer 6'\n 'go to drawer
↪   7'\n 'go to drawer 8'\n 'go to dresser 1'\n 'go to garbagecan 1'\n
↪   'go to shelf 1'\n 'go to shelf 2'\n 'go to shelf 3'\n 'go to shelf
↪   4'\n 'go to shelf 5'\n 'inventory'\n 'look'\n 'move pencil 2 to
↪   desk 1'

memory: "Looking at the past observations and actions from our
↪   conversation history, the most relevant memory for this step is:
↪   \nAt step 2, the observation was 'You arrive at desk 1. On the desk
↪   1, you see a book 2, a cd 2, a creditcard 2, and a pencil 2.' and
↪   the action was 'take pencil 2 from desk 1'. This memory is helpful
↪   because it indicates that there is at least one pencil on the desk,
↪   and we have already taken it. Now, we need to find another pencil
↪   and put both pencils in a drawer."

reflection: "Reflecting on the last action and its outcome, we can see
↪   that the last action 'take pencil 2 from desk 1' was successful,
↪   and we have picked up the pencil 2 from the desk 1. We have made
↪   progress toward the task goal, but we still need to find another
↪   pencil and put both pencils in a drawer. The task is not yet
↪   complete."
plan: "Planning the next step based on memory and reflection, we should
↪   examine the desk 1 again to see if there is another pencil. This
↪   plan is helpful because it allows us to gather more information
↪   about the objects on the desk and potentially find the second
↪   pencil. We expect this action to achieve a better understanding of
↪   the objects on the desk and possibly find the second pencil."
action: "examine desk 1"
```

Figure 23: **Error Type: Inefficient Plan**

**Error Type - Misalignment**

| | |
|---|---|
| **task type** | GAIA |
| **task description** | |
| **critical failure module** | action |
| **evidence** | Evidence: Plan: 'I will modify the approach to directly explore the Tropicos database or a reliable botanical taxonomy reference to find the Tropicos ID for the Order Helotiales.' Action: tool: wikipedia knowledge searcher parameters: "query": "Helotiales Tropicos ID" |
| **critical step** | 3 |

**step trajectory**

```
observation: Tool 'google_search' executed.\nResult: [\n  {\n
↪   \"title\": \"Untitled\",\n    \"link\":
↪   \"https://www.tropicos.org/name/Search?name=Helodiaceae\",\n
↪   \"snippet\": \"\"\n  }\n]

memory: "In step 1, I attempted to find the Tropicos ID for the Order
↪   Helotiales using a Google search, which resulted in a link that was
↪   not directly relevant.\n- In step 2, I used the Wikipedia knowledge
↪   searcher to find the Tropicos ID for Helotiales, but no results
↪   were found for the query."

reflection: "The last action did not yield the desired Tropicos ID for
↪   the Order Helotiales, as no results were found on Wikipedia. This
↪   indicates that Wikipedia might not have the specific identifier
↪   information needed, and it confirms the need to access more
↪   specialized or direct resources for taxonomic data. Although there
↪   has been no progress in finding the Tropicos ID, identifying the
↪   limitations of current methods is valuable for refining the search
↪   approach."

plan: "I will use the Google search tool again, but with a refined
↪   query to specifically target the Tropicos database or another
↪   authoritative botanical taxonomy site that might list the Tropicos
↪   ID for Helotiales. This plan is likely to be effective because it
↪   focuses on accessing specialized databases where the Tropicos ID is
↪   more likely to be documented. The goal of this action is to obtain
↪   the Tropicos ID so that I can proceed with calculating the ISBN-10
↪   check digit."
action: "tool: google_search\nparameters: {\"query\": \"Helotiales
↪   Tropicos ID site:tropicos.org\", \"num_results\": 1}"
```

Figure 24: **Error Type: Misalignment**

---

**Error Type - Format Error**

| | |
|---|---|
| **task type** | alfworld |
| **task description** | clean some plate and put it in shelf. |
| **critical failure module** | action |
| **evidence** | Step 15 has a non-empty plan but the recorded action is empty. Step 16 observation returns 'Nothing happens'. From Step 17 onward, admissible actions at the dining table never include 'take plate 1 ...', yet attempts persist. |
| **critical step** | 15 |

**step trajectory**

```
observations: "On the diningtable 1, you see a apple 1, a bread 1, a
↪  butterknife 1, a cup 3, a fork 1, a houseplant 1, a knife 1, a
↪  lettuce 3, a lettuce 2, a peppershaker 2, a plate 2, a plate 1, a
↪  saltshaker 1, a soapbottle 2, a soapbottle 1, a spoon 2, a spoon 1,
↪  a winebottle 3, and a winebottle 2."

admissible_actions: "'examine diningtable 1'\n 'examine dishsponge 1'\n
↪  'go to cabinet 1'\n 'go to cabinet 10'\n 'go to cabinet 11'\n 'go
↪  to cabinet 12'\n 'go to cabinet 13'\n 'go to cabinet 2'\n 'go to
↪  cabinet 3'\n 'go to cabinet 4'\n 'go to cabinet 5'\n 'go to cabinet
↪  6'\n 'go to cabinet 7'\n 'go to cabinet 8'\n 'go to cabinet 9'\n
↪  'go to coffeemachine 1'\n 'go to countertop 1'\n 'go to countertop
↪  2'\n 'go to drawer 1'\n ..."

memory: "Looking at the past observations and actions from our
↪  conversation history, a relevant memory for this step is from step
↪  10, where the observation was \"You open the drawer 1. The drawer 1
↪  is open. In it, you see a dishsponge 1.\" and the action was \"take
↪  dishsponge 1 from drawer 1\". This memory is helpful because it
↪  shows that the dishsponge 1, which is likely needed for cleaning,
↪  has been retrieved and is now in the agent's possession. Another
↪  relevant memory is from step 14, where the observation was ..."

reflection: "Reflecting on the last action and its outcome, the last
↪  action was \"examine diningtable 1\" and the outcome was that the
↪  agent observed the current state of the diningtable 1. The task
↪  goal is to clean some plate and put it in a shelf, and the last
↪  action did not directly contribute to completing this goal.
↪  However, it did provide information about the current state of the
↪  diningtable 1, which is necessary for planning the next step. The
↪  agent is making progress toward the task goal by gathering
↪  information, but the actual cleaning and putting away of the plate
↪  has not started yet. To make progress, the agent should use the
↪  dishsponge 1 to clean a plate and then put it away in a shelf.",

plan: "Planning the next step based on memory and reflection, the agent
↪  should use the dishsponge 1 to clean a plate. Since the agent is
↪  currently at the diningtable 1 and there are plates present, the
↪  agent should take a plate from the diningtable 1 and clean it using
↪  the dishsponge 1. After cleaning the plate, the agent can put it
↪  away in a shelf. This plan is helpful because it directly addresses
↪  the task goal and uses the information gathered from previous
↪  actions. The expected outcome of this action is that the agent will
↪  have a clean plate that can be put away in a shelf."

action: "take plate 1 from diningtable 1."
```

Figure 25: **Error Type: Format Error**

---

**Error Type - Progress Misjudge**

| | |
|---|---|
| **task type** | alfworld |
| **task description** | put a clean bowl in shelf. |
| **critical failure module** | reflection |
| **evidence** | Step 16 reflection: 'This means that the task goal of putting a clean bowl in the shelf has been completed. The last action was successful, and progress was made toward the task goal. However, to confirm that the bowl is indeed in the shelf and the task is fully completed, further verification might be necessary.' |
| **critical step** | 16 |

**step trajectory**

```
observations: ""You move the bowl 1 to the shelf 1."

admissible_actions: "'examine shelf 1'\n 'go to cabinet 1'\n 'go to
↪  cabinet 2'\n 'go to cabinet 3'\n 'go to cabinet 4'\n 'go to cabinet
↪  5'\n 'go to cabinet 6'\n 'go to cabinet 7'\n 'go to cabinet 8'\n
↪  'go to cabinet 9'\n 'go to coffeemachine 1'\n 'go to countertop
↪  1'\n 'go to countertop 2'\n 'go to countertop 3'\n 'go to drawer
↪  1'\n 'go to drawer 2'\n 'go to drawer 3'\n 'go to drawer 4'\n 'go
↪  to drawer 5'\n 'go to drawer 6'\n 'go to drawer 7'\n 'go to drawer
↪  8'\n 'go to drawer 9'\n 'go to fridge 1'\n 'go to garbagecan 1'\n
↪  'go to microwave 1'\n 'go to shelf 2'\n 'go to shelf 3'\n 'go to
↪  sinkbasin 1'\n 'go to stoveburner 1'\n 'go to stoveburner 2'\n 'go
↪  to stoveburner 3'\n 'go to stoveburner 4'\n 'go to toaster 1'\n
↪  'inventory'\n 'look'\n 'take bowl 1 from shelf 1'."

memory: "Looking at the past observations and actions from our
↪  conversation history, a relevant memory for this step is from Step
↪  13 and Step 14. \nAt Step 13, the observation was: \"You arrive at
↪  countertop 1. On the countertop 1, you see a bowl 1, a butterknife
↪  2, a butterknife 1, a dishsponge 1, a fork 1, and a kettle 1.\"
↪  \nThe action taken was: \"take bowl 1 from countertop 1\". \nThen
↪  at Step 14, the observation was: \"You pick up the bowl 1 from the
↪  countertop 1.\" \nThe action taken was: \"go to shelf 1\". \nThis
↪  memory is helpful because it shows how the bowl was obtained and
↪  moved to the shelf, which is the goal of the task."

reflection: "Reflecting on the last action and its outcome, the last
↪  action was to move the bowl 1 to the shelf 1. \nThe outcome was
↪  that the bowl was successfully moved to the shelf. \nThis means
↪  that the task goal of putting a clean bowl in the shelf has been
↪  completed. \nThe last action was successful, and progress was made
↪  toward the task goal. \nHowever, to confirm that the bowl is indeed
↪  in the shelf and the task is fully completed, further verification
↪  might be necessary.",

plan: "Planning the next step based on memory and reflection, given
↪  what has been learned, the next action should be to verify that the
↪  bowl is indeed on the shelf. \nThis plan is helpful because it
↪  ensures that the task goal has been fully achieved and that there
↪  are no unexpected issues. \nThe expected outcome of this action is
↪  to confirm the presence of the bowl on the shelf, thereby
↪  confirming the completion of the task."

action: "examine shelf 1."
```

Figure 26: **Error Type: Progress Misjudge**

