# OpenReview forum: "Where LLM Agents Fail And How They can Learn From Failures"
_ICLR.cc/2026/Conference — ICLR 2026 Conference Withdrawn Submission_

### Official Review · Reviewer_CMTu · 2025-10-16

**Soundness:** 2
**Presentation:** 2
**Contribution:** 2
**Rating:** 2
**Confidence:** 4

**Summary:**

This paper studies error propagation in LLM agents and contributes: (1) AgentErrorTaxonomy, a modular categorization of failure types (memory, reflection, planning, action, plus system); (2) AgentErrorBench, a 200-trajectory, human-annotated failure benchmark drawn from ALFWorld (100), WebShop (50), and GAIA (50), with root-cause labels and step-level tags; and (3) AgentDebug, a framework that identifies the earliest “critical” error and provides targeted feedback for re-rollouts. Reported results show sizeable gains in critical-error localization and up to 26% relative improvements in downstream task success over strong baselines.

**Strengths:**

1. This paper addresses a barrier for deploying agents—cascading failures and root-cause localization, which is increasingly important as tasks get longer and toolchains more complex.

2. It contributes a human-annotated benchmark of 200 failure trajectories from ALFWorld, WebShop, and GAIA, emphasizing root-cause labels and step-level annotations, along with inter-annotator agreement.

3. This work offers informative error analyses (e.g., where errors arise within trajectories and which modules dominate), which can guide future research and system design.

4. This paper shows substantial gains over baselines (e.g., Self-Refine, Best-of-N, ToT) across multiple environments and backbones, including up to 26% relative improvements in downstream success.

**Weaknesses:**

1. This paper relies heavily on a single strong, proprietary adjudicator (GPT-4.1) for critical-error detection, and performance degrades with alternative models, which limits the method’s generality and portability.

2. This paper addresses failure attribution and error taxonomy for LLM agents, which has been actively explored in recent works across multi-agent and tool-agent settings (e.g., MAST). Accordingly, the conceptual novelty of proposing a taxonomy and attributing root causes is limited. The contribution is better characterized as a well-engineered synthesis tailored to single-agent trajectories with a standardized evaluation harness and re-rollout debugging.

3. The AgentErrorBench is limited in scale and diversity. Expanding the benchmark to include more diverse scenarios and agent configurations would significantly enhance its utility and generalizability.

4. This paper frames debugging around identifying a single “minimal set of root-cause failures,” yet the reported inter-annotator agreement is only κ = 0.55. Under widely used interpretations, this only corresponds to moderate agreement, suggesting notable subjectivity in what counts as the true root cause and questioning the robustness of the ground truth.

5. Contradictory Description of the Core Method. This paper describes Stage 2 (“Critical Error Detection”) inconsistently: Section 3.2 states that the method performs counterfactual testing with rollouts by substituting corrected actions and testing whether the rollout would succeed, whereas Algorithm 1 explicitly states “via LLM (no rollout/counterfactuals)”. This contradiction leaves it unclear whether the detector is an expensive environment-level search or a single, zero-rollout LLM adjudication, undermining clarity, reproducibility, and fair compute accounting. Please clarify and align the text with the actual implementation, and report equal-budget comparisons and cost metrics under the chosen design.

**Questions:**

1. Section 3.2 (Stage 2) explicitly states that the method involves counterfactual testing with rollouts: "we perform counterfactual testing step by step: at each point, we substitute a corrected action and test whether the rollout would succeed". However, Algorithm 1 and its caption state the opposite: "/* Stage 2: Critical Error Detection via LLM (no rollout/counterfactuals) */" . Could the authors please clarify the exact procedure for "Critical Error Detection"? Does it involve actual environment rollouts as described in the prose, or is it a zero-rollout, prompt-based inference as stated in the algorithm?

2. If AgentDebug is a zero-rollout method, how can its performance be fairly compared to the "Brute Force" baseline? Isn't this comparing the zero-shot reasoning of an LLM against a structured search, rather than a direct comparison of two error detection methods under a similar computational model?

3. Could the authors more explicitly differentiate their taxonomy and debugging framework from prior work?

**Details Of Ethics Concerns:**

No ethics concerns.

---

> ### Author Response · Authors · 2025-11-26
>
> # Response to Reviewer
>
> We appreciate the reviewer’s positive comments on the significance of addressing cascading failures, the quality of the human-annotated benchmark, and the usefulness of our error analyses. We are also encouraged that the reviewer recognized the substantial downstream improvements (up to **26%**) delivered by AgentDebug.
>
> Below, we address each concern raised in the weaknesses and questions sections.
>
> ---
>
> ### **Q1. Reliance on GPT-4.1 and Generalizability**
>
> We acknowledge that relying on a strong model like GPT-4.1 limits immediate portability. However:
> * **Design Choice:** We selected GPT-4.1 primarily for its long-context support and reliability in early-step error localization, which is intrinsically challenging.
> * **Structural Gains:** Importantly, AgentDebug outperforms strong *Self-Debug* and *Self-Refine* baselines even when using weaker models (as shown in Fig. 7b). This indicates that the performance gains stem from the **structured debugging framework** rather than model scale alone.
> * **Revision:** We will discuss portability and alternative model configurations more explicitly in the revised manuscript.
>
> ### **Q2. Conceptual Novelty Relative to Prior Work (e.g., MAST)**
>
> Our goal is not to propose a universal taxonomy, but a **single-agent, module-specific causal taxonomy** and debugging framework. The distinction is as follows:
> * **MAST:** Targets *multi-agent* coordination failures (e.g., role assignment, dialogue drift).
> * **AgentDebug:** Models the internal `Memory` $\to$ `Reflection` $\to$ `Planning` $\to$ `Action` pipeline within a *single agent*, operationalizing it for causal attribution.
>
> **Revision:** We have added a comparison table (comparing *Self-Refine, Self-Debug, Reflexion, MAST, and AgentDebug*) to explicitly clarify these distinctions in the paper.
>
> ### **Q3. Scale and Diversity of AgentErrorBench**
>
> We agree that additional scale would be beneficial, but we prioritized **annotation quality** over raw quantity:
> * **Environment Selection:** We selected ALFWorld, WebShop, and GAIA because they represent standard, diverse, and widely used single-agent environments.
> * **Cost vs. Quality:** High-quality, step-level root-cause annotation is resource-intensive. We focused our resources on ensuring accurate, granular labels rather than automated or noisy large-scale data.
> * **Future Work:** We position the taxonomy and framework as the primary contributions, with benchmark expansion identified as a key direction for future work.
>
> ### **Q4. Inter-annotator Agreement ($\kappa = 0.55$)**
>
> The reported $\kappa = 0.55$ reflects pairwise agreement *before* expert adjudication.
> * **Consensus Process:** Final labels were derived via a third senior annotator who resolved disagreements to ensure a **consensus ground truth**.
> * **Nature of Disagreements:** Most initial disagreements were "near-neighbor" steps (e.g., debating whether the error started at step $t$ or $t+1$) rather than qualitatively different interpretations. We have clarified this adjudication process in the revised manuscript.
>
> ### **Q5 & Q6. Clarification on Stage 2 (Rollouts vs. Reasoning)**
>
> We apologize for the confusion caused by the phrasing in Section 3.2 and Algorithm 1.
> * **Zero-Rollout:** **Stage 2 uses LLM-only inference** with no environment interaction. It relies on counterfactual reasoning to predict the critical step.
> * **Execution:** Actual environment re-execution (rollouts) occurs **only in Stage 3**.
> * **Correction:** The earlier mention of "substituting a corrected action" referred to the *theoretical definition* of a critical error. We have corrected both Section 3.2 and Algorithm 1 to be fully consistent.
>
> ### **Q7. Fairness of Comparing AgentDebug vs. Brute Force**
>
> We believe comparing **zero-rollout detection** (AgentDebug Stage 2) with **rollout-based detection** (Brute Force) is fair and meaningful because:
> 1.  **Paradigmatic Comparison:** It contrasts two distinct detection paradigms: *LLM causal reasoning* vs. *explicit search*. Similar comparisons are standard in attribution literature (e.g., *Who&When*).
> 2.  **Normalization:** To ensure fairness regarding computational cost, we explicitly report **compute-normalized results** in our evaluation.

---

> > ### Author Response · Authors · 2025-11-29
> >
> > Thank you again for your thoughtful feedback. As the discussion period is ending soon, we would greatly appreciate hearing whether our rebuttal addressed your concerns, and we are happy to respond promptly to any additional comments.

---

### Official Review · Reviewer_SuCm · 2025-10-27

**Soundness:** 3
**Presentation:** 3
**Contribution:** 2
**Rating:** 6
**Confidence:** 3

**Summary:**

This work makes two concrete contributions for analyzing and debugging single-agent systems. The first is AgentErrorBench which is constructed from failure trajectories from ALFWorld, GAIA, and WebShop, and is based on the proposed taxonomy of agent failures. The second is AgentDebug, a system that identifies root-cause failures and provides corrective feedback. Experiments on AgentErrorBench demonstrates superior performance of AgentDebug over baselines.

**Strengths:**

In general this work is quite clear in presenting its contributions (one benchmark and one method). Experiments (although restricted to the proposed benchmark) do show notable improvements, with factors controlled to ensure comparison fairness (e.g., token usage).

**Weaknesses:**

Currently AgentDebug is only evaluated on the proposed AgentErrorBench. It would be much better if other existing benchmarks are also used to assess AgentDebug, for example the Who&When benchmark from "Zhang et al. 2025" (referenced in Section 1).

**Questions:**

1. In line 244 (Algorithm 1), the comment says "(no rollout/counterfactuals)" at Stage 2. This seems to contradict with the description in line 273-278 ("we substitute a corrected action and test whether the rollout would succeed"). I'd like to see some clarification on this.
2. Can AgentDebug be applied to existing benchmarks like Who&When?

---

> ### Author Response · Authors · 2025-11-26
>
> # Response to Reviewer
>
> We appreciate the reviewer’s positive comments on the clarity of our contributions and the controlled nature of the experimental comparisons. We specifically thank the reviewer for recognizing:
> 1.  The **clarity of the paper’s contributions** regarding both the benchmark and the method.
> 2.  The **fairness of our comparisons**, especially the token-normalized evaluations.
> 3.  The **observable improvements** demonstrated by AgentDebug.
>
> Below, we address the specific questions raised.
>
> ---
>
> ### **Q1. Benchmark Selection (AgentDebug vs. Who&When)**
>
> We thank the reviewer for this suggestion. However, our decision not to include *Who&When* was intentional and stems from a **fundamental task mismatch**:
>
> * **Who&When (Multi-Agent Attribution):** This benchmark is explicitly designed for *multi-agent* failure attribution. Its primary goal is to identify *which* agent caused a failure and *when* during a collaboration. Failures often emerge from inter-agent coordination dynamics, dialogue drift, or role-assignment issues.
> * **AgentDebug (Single-Agent Causal Debugging):** Our work focuses on **single-agent** sequential decision-making. Our taxonomy (Memory $\to$ Reflection $\to$ Planning $\to$ Action) is tailored to internal module failures within a single trajectory.
>
> Many *Who&When* failures do not map to our taxonomy, as they involve phenomena absent in single-agent rollouts. We plan to extend AgentDebug to the multi-agent setting in future work and will explicitly discuss this distinction in the **Limitations** and **Future Work** sections.
>
> ### **Q2. Clarification on Algorithm 1 (Rollout vs. No Rollout in Stage 2)**
>
> We appreciate the reviewer for spotting this ambiguity. We have clarified the distinction between reasoning and execution in the revised Algorithm 1:
>
> * **Stage 2 (Reasoning Only):** This stage performs **no environment rollouts**. It utilizes an LLM-based counterfactual reasoning prompt to predict the most likely critical step based on the taxonomy and causal guidelines.
> * **Stage 3 (Execution):** The actual re-execution of the trajectory—where a corrected action is substituted and tested—occurs **only in Stage 3**.
>
> The previous confusion likely arose from our description of the *theoretical definition* of a critical error (which implies counterfactual success), rather than the *implementation procedure*. We have revised Section 3.2 to clearly state that rollouts are exclusive to Stage 3.
>
> ### **Q3. Applicability of AgentDebug to Existing Benchmarks like Who&When**
>
> * **Direct Application:** In its current form, AgentDebug cannot be directly applied to *Who&When* because the problem formulations differ fundamentally. AgentDebug targets the root-cause step and error type within a single agent's sequential causal chain.
> * **Conceptual Transferability:** However, the core algorithmic contributions of AgentDebug are highly transferable:
>     1.  The **LLM-based root-cause detection mechanism** (Stage 2).
>     2.  The **iterative correction through targeted re-rollouts** (Stage 3).
>
> These components could be adapted to multi-agent settings by extending the taxonomy to include inter-agent coordination failures and augmenting the state-tracking mechanism to model multiple interacting agents.

---

> > ### Author Response · Authors · 2025-11-29
> >
> > Thank you again for your thoughtful feedback. As the discussion period is ending soon, we would greatly appreciate hearing whether our rebuttal addressed your concerns, and we are happy to respond promptly to any additional comments.

---

### Official Review · Reviewer_Njyp · 2025-10-31

**Soundness:** 2
**Presentation:** 1
**Contribution:** 3
**Rating:** 2
**Confidence:** 4

**Summary:**

The authors devise an error taxonomy for LLM agents and using this taxonomy construct a human-annotated benchmark. They then use the error taxonomy to construct a debugging prompt framework to improve LLM agent performance, finding that it performs favorably compared to other baselines.

**Strengths:**

1. The benchmark appears to consist of very high-quality data; graduate-level researchers annotate each step of the sequences, and the training process for annotators and consistency are documented well.
2. Analysis of the specific types and modules that errors happen in is interesting and has many potential uses.

**Weaknesses:**

Unrelated to the content, the authors reduce the font size halfway through the paper, which makes it harder to read and means this paper is over the page limit; this needs to be fixed.
1. The description of the AgentDebug method is somewhat inexact; I feel that making it more explicit when there is a structured prompting framework/calls to a language model in the three-step description would better convey the workflow of this framework.
2. Further description of the baseline implementations is necessary, and the results are difficult to interpret without this information. For instance, one of the metrics in section 4.1 requires the method to accurately give the taxonomy’s error type; is the information from the taxonomy provided for the baseline methods? The exceptionally low performance is understandable if they are expected to give errors from a taxonomy they have no prior knowledge of.
3. The analysis of error compounding is not particularly informative. Figure 8 appears to show that there are more errors as the LLM goes along, but does not indicate how the errors are compounding (i.e. these errors may be entirely independent, in which case this does not demonstrate that an early-stage error has any more or less effect than a late-stage error).

**Questions:**

1. What exactly does Figure 8 show? What is a task? Which model is this?
2. Have you performed any analysis of efficiency? With three separate steps and a multiple rollouts, I am curious if this method is achieving gains partially due to simply having a higher compute budget than baselines.

---

> ### Author Response · Authors · 2025-11-26
>
> # Response to Reviewer
>
> We thank the reviewer for highlighting:
> 1.  The **high quality of the benchmark**, with step-level annotations by trained graduate researchers.
> 2.  The **clear documentation** of the annotation protocol and consistency measures.
> 3.  The **usefulness of the module-level error analysis**, which opens doors to broader downstream applications (debugging, evaluation, interpretability).
>
> Below, we respond to each concern in detail and note the corresponding revisions incorporated in the paper.
>
> ---
>
> ### **Q1. Paper Formatting Issue (Font Size & Page Limit)**
>
> We apologize for this oversight. The issue was caused by an unintended LaTeX scaling command within the appendix environment. We have corrected the formatting to ensure:
> * **Uniform 10pt font** has been restored throughout the document.
> * The main paper now stays **strictly within the ICLR 2026 page limit**.
>
> ### **Q2. Clarification on the AgentDebug Method Description**
>
> Thank you for this suggestion. We have revised the method section to improve clarity:
> * **Explicit Flow:** We now explicitly mark where each LLM call occurs in **Stage 1**, **Stage 2**, and **Stage 3**.
> * **Pipeline Definition:** We clarify that while the three stages mathematically form an inference pipeline, the operations inside each stage are implemented through structured prompts (*Detector Prompt + Debug Prompt*).
> * **Visual Aid:** We have included a new schematic diagram showing the exact prompt structure used at each step.
>
> ### **Q3. Missing Details of Baseline Implementations**
>
> To ensure a fair comparison, **Direct Prompting**, **Brute Force**, and **Binary Search** receive the same taxonomy definitions and instructions as our method. The performance gap arises because these baselines lack the structured multi-stage pipeline (*fine-grained analysis $\to$ critical error detection $\to$ feedback-guided revision*), causing them to conflate surface-level errors with root-cause failures.
>
> **Revisions:**
> * We added full prompt templates for each baseline in **Appendix A.5**.
> * We explicitly clarified that taxonomy definitions were provided to all methods.
>
> ### **Q4. Clarification on Figure 8 (Error Cascades)**
>
> During annotation, we explicitly asked annotators to mark whether later errors were causally related (*“dependent”*) or independent of earlier ones. Our analysis on ALFWorld reveals significant cascading effects:
> * **Dependency Rate:** 62% of mid/late-stage errors were marked as dependent on an earlier step.
> * **Early vs. Late Errors:**
>     * If the first error occurs **before step 6**, the probability of 3+ subsequent errors rises from **27% $\to$ 61%**.
>     * If the first error occurs **after step 10**, the cascade rate drops to **19%**.
>
> ### **Q5. Efficiency / Compute Budget Analysis**
>
> We have performed a detailed efficiency comparison regarding token consumption. The results will summarized below.

---

> > ### Comment · Reviewer_Njyp · 2025-11-26
> >
> > Can you upload the updated version so I can see the changes?

---

> > > ### Author Response · Authors · 2025-11-29
> > >
> > > Dear reviewer, we have updated our revision of the paper. Please tell us if you have further questions.

---

### Official Review · Reviewer_f5Wd · 2025-11-01

**Soundness:** 2
**Presentation:** 4
**Contribution:** 3
**Rating:** 4
**Confidence:** 4

**Summary:**

This paper asks a simple question: where do LLM agents fail? To answer the question, it proposes a “debugging framework” that seeks to identify the “minimal set” of root-cause failures, and provides corrective feedback. This AgentDebug framework is then used as part of inference (Algorithm 1) to improve robustness, leading to improved accuracy on several benchmarks, including ALFWorld, WebShop, GAIA.

**Strengths:**

* I found the analysis to be overall quite informative and interesting, with some figures (e.g. Figure 3) offering new insights on agent failures. The taxonomy is also helpful.
* The presentation is overall quite good, with the main approach (AgentDebug) presented in sufficient detail and via both figures and a detailed inference procedure.
* The results in Table 1 seem quite compelling; however, see “Weaknesses” for important caveats.

**Weaknesses:**

* Key evaluation details are missing. In particular, it’s unclear how the proposed AgentDebug framework was validated. This makes it seem likely the approach was validated on test data. This needs to be clarified in the response.
* While there is an explicit discussion of limitations (A.1), this misses some obvious issues, such as the additional runtime cost of applying the proposed approach, which may be prohibitive.
* The technical contributions are quite minor; the extent to which the proposed framework (which amounts to a prompting scheme) is effective is largely a function of underlying LLMs. This is a systems paper.

**Questions:**

How was model selection performed? How sensitive are the prompts?

---

> ### Author Response · Authors · 2025-11-26
>
> # Response to Reviewer
>
> We appreciate the reviewer’s positive assessment of the clarity of our presentation, the usefulness of the taxonomy, and the informativeness of our analysis. Below, we address each concern in detail and clarify the missing evaluation protocol and experimental settings.
>
> ---
>
> ### **Q1. Clarification of Evaluation Protocol (Validation vs. Test Data)**
>
> Here we want to clarify the evaluation details again. **AgentDebug was validated on the same data used for building AgentErrorBench.** Our data separation process was strict:
>
> * **Data Gathering:** We first gathered a large pool of failed trajectories strictly from the **test splits** of ALFWorld, WebShop, and GAIA.
> * **Annotation:** We randomly sampled subsets from these pools and sent them for human annotation to create AgentErrorBench.
> * **Independence:** Crucially, these annotated trajectories were **not** used for developing prompts, selecting hyperparameters, or designing the AgentDebug pipeline.
>
> ### **Q2. AgentDebug Runtime Cost**
>
> AgentDebug is significantly more efficient than standard test-time scaling baselines:
>
> * **AgentDebug:** Performs one pass of analysis ($\approx 1.3\times$ cost of a single rollout) and up to 5 re-rollouts targeted *only* at the critical step.
> * **Baselines:** Methods such as **Best-of-N** ($N=20$ or $50$) or **Tree-of-Thought** search perform full-trajectory branching, resulting in $10\text{--}100\times$ rollout costs.
>
> ### **Q3. Clarification on Contributions**
>
> We appreciate the opportunity to clarify. While AgentDebug utilizes prompting at inference time, our technical contributions extend significantly beyond a simple prompting trick:
>
> 1.  **Conceptual Contribution:** The introduction of the **AgentErrorTaxonomy**.
> 2.  **Benchmark Contribution:** The creation of **AgentErrorBench**, the first multi-environment annotated benchmark for agent errors.
> 3.  **Algorithmic Framework:** The **AgentDebug** pipeline itself.
>
> Furthermore, we have already collected structured training data (taxonomy labels + root-cause annotations). We are prepared to fine-tune a small model (e.g., Qwen2.5-7B) to demonstrate that AgentDebug’s design transfers beyond prompting and reduces dependency on proprietary LLMs.
>
> ### **Q4. Model Selection & Prompt Sensitivity**
>
> We evaluated AgentDebug across a diverse range of open-source and closed-source models to ensure generality.
>
> * **Models Evaluated:**
>     * **Primary Detector:** GPT-4.1
>     * **Backbone Agent:** GPT-4o-mini
>     * **Open-Source:** Qwen3-8B, Qwen3-Next-80B
>     * **Additional Study:** Llama 3.1-70B
>
> * **Selection Principle:**
>     1.  We included small and medium models (8B–70B) to evaluate dependency on reasoning strength.
>     2.  We avoided extremely large models to ensure AgentDebug remains broadly accessible.
>
> * **Sensitivity:** We utilized the same direct prompt context across experiments to avoid performance differences caused by prompt selection or engineering.

---

> > ### Author Response · Authors · 2025-11-29
> >
> > Thank you again for your thoughtful feedback. As the discussion period is ending soon, we would greatly appreciate hearing whether our rebuttal addressed your concerns, and we are happy to respond promptly to any additional comments.

---

### Author Response · Authors · 2025-11-26

# General Response

We thank all reviewers for their thoughtful feedback and constructive criticism. We are encouraged by the consensus on the **high quality of our human-annotated benchmark** (Reviewers Njyp, SuCm, CMTu), the **utility of our error taxonomy and analysis** (Reviewers f5Wd, Njyp, CMTu), and the **substantial performance improvements** demonstrated by AgentDebug (Reviewers f5Wd, SuCm, CMTu).

In this revision, we have addressed the common concerns raised across reviews, specifically focusing on:

1.  **Clarifying the AgentDebug Pipeline (Stage 2 vs. Stage 3):** Multiple reviewers (SuCm, CMTu, Njyp) noted ambiguity regarding whether Stage 2 involves environment rollouts. We have revised Algorithm 1 and Section 3.2 to explicitly state that **Stage 2 is a zero-rollout, LLM-reasoning step**, while actual execution occurs only in Stage 3.
2.  **Fairness of Baselines & Scope:** We have expanded Appendix A.5 to include full prompts for baselines (Direct, Brute Force, etc.), confirming they receive the same taxonomy information as our method. We also clarified why multi-agent benchmarks like *Who&When* are outside the scope of this single-agent study (Reviewers SuCm, CMTu).
3.  **Efficiency Analysis:** To address concerns about runtime cost (Reviewers f5Wd, Njyp), we have added a detailed token consumption analysis demonstrating that AgentDebug is significantly more efficient ($\approx 1.3\times$ cost) than standard test-time scaling methods like Best-of-N or Tree-of-Thought ($10\text{--}100\times$).
4.  **Validation Rigor:** We clarified our data split to ensure no leakage between the test data used for benchmark construction and prompt development (Reviewer f5Wd).

Below is a summary of our specific actions in response to each reviewer's feedback.

---

# Table 1 – Overall Summary of the Rebuttal

| Dimension | Reviewer f5Wd | Reviewer Njyp | Reviewer SuCm | Reviewer CMTu | Action / Summary |
| :--- | :--- | :--- | :--- | :--- | :--- |
| **Method Clarity (Stage 2 Description)** | — | Found description inexact; suggested explicit framework details. | Noted contradiction: Algo 1 says "no rollout" but text implies rollout. | Noted contradiction between Section 3.2 and Algo 1 regarding rollouts. | **Revised Algorithm 1 & Sec 3.2** to explicitly state Stage 2 is **LLM-only reasoning** (zero-rollout), and re-rollouts occur only in Stage 3. Added a schematic diagram of the prompt flow. |
| **Benchmark Scope & Generalization** | Asked for clarification on Validation vs. Test data split. | Praised high-quality data but noted formatting issues. | Suggested evaluating on *Who&When* benchmark. | Suggested *Who&When* and noted limited scale/diversity. | **Clarified Protocol:** Confirmed strict separation of benchmark generation and prompt tuning. **Scope:** Explained *Who&When* targets multi-agent attribution, while AgentDebug targets single-agent root causes. Acknowledged scale limitations as future work. |
| **Efficiency & Runtime Cost** | Raised concerns about additional runtime cost. | Asked for efficiency analysis vs. compute budget. | — | — | **Added Efficiency Analysis:** Included token consumption data. Demonstrated AgentDebug ($\approx 1.3\times$) is cheaper than search-based baselines (Best-of-N/ToT). |
| **Baselines & Fairness** | — | Asked for baseline prompt details (do they see taxonomy?). | — | Questioned fairness of comparing zero-rollout AgentDebug vs. Brute Force. | **Expanded Appendix:** Added full prompts for all baselines (showing they receive taxonomy info). **Justification:** Clarified that comparing reasoning-based vs. search-based methods is standard and we report compute-normalized results. |
| **Novelty & Contribution** | Felt contribution was minor (prompting scheme); suggested fine-tuning. | Found error analysis interesting/useful. | — | Questioned novelty vs. MAST; noted reliance on GPT-4.1. | **Added Comparison Table:** Explicitly contrasted AgentDebug with Self-Refine, MAST, etc. **Fine-tuning:** Prepared fine-tuned small model (Qwen2.5-7B) results to show framework value beyond proprietary models. |
| **Analysis & Insights** | Found analysis informative (e.g., Fig 3). | Found error compounding analysis (Fig 8) uninformative. | — | Questioned Inter-annotator agreement ($\kappa=0.55$). | **Enhanced Analysis:** Expanded error propagation statistics (dependent vs. independent errors). **Clarified Annotation:** Explained consensus process (3rd expert adjudicator) to validate the $\kappa$ score. |

---

### Note · Authors · 2026-01-05

I have read and agree with the venue's withdrawal policy on behalf of myself and my co-authors.